# Single-pass Streaming Lower Bounds for Multi-armed Bandits Exploration with Instance-sensitive Sample Complexity

**Sepehr Assadi** [*]
Department of Computer Science
Rutgers University
Piscataway, NJ 08854
sepehr.assadi@rutgers.edu

**Chen Wang** [*]
Department of Computer Science
Rutgers University
Piscataway, NJ 08854
chen.wang.cs@rutgers.edu

## Abstract

Motivated by applications to process massive datasets, we study streaming algorithms for pure exploration in Stochastic Multi Armed Bandits (MABs). This problem was first formulated by Assadi and Wang [STOC 2020] as follows: A collection of $n$ arms with unknown rewards are arriving one by one in a stream, and the algorithm is only allowed to store a limited number of arms at any point. The goal is to find the arm with the largest reward while minimizing the number of arm pulls (sample complexity) and the maximum number of stored arms (space complexity). Assuming $\Delta_{[2]}$ is known, Assadi and Wang designed an algorithm that uses a memory of just one arm and still achieves the sample complexity of $O(n/\Delta_{[2]}^2)$ which is *worst-case* optimal even for non-streaming algorithms; here $\Delta_{[i]}$ is the gap between the rewards of the best and the $i$-th best arms.

In this paper, we extended this line of work to stochastic MABs in the streaming model with the *instance-sensitive* sample complexity, i.e. the sample complexity of $O(\sum_{i=2}^{n} \frac{1}{\Delta_{[i]}^2} \log\log{(\frac{1}{\Delta_{[i]}})})$, similar in spirit to Karnin et.al. [ICML 2013] and Jamieson et.al. [COLT 2014] in the classical setting. We devise strong negative results under this setting: our results show that any streaming algorithm under a single pass has to use either asymptotically higher sample complexity than the instance-sensitive bound, or a memory of $\Omega(n)$ arms, even if the parameter $\Delta_{[2]}$ is known. In fact, the lower bound holds under much stronger assumptions, including the random order streams or the knowledge of all gap parameters $\{\Delta_{[i]}\}_{i=2}^{n}$. We complement our lower bounds by proposing a new algorithm that uses a memory of a single arm and achieves the instance-optimal sample complexity when all the strong assumptions hold simultaneously.

Our results are developed based on a novel *arm trapping lemma*. This generic complexity result shows that any algorithm to *trap* the index of the best arm among $o(n)$ indices (but not necessarily to *find* it) has to use $\Theta(n/\Delta_{[2]}^2)$ sample complexity. This result is *not* restricted to the streaming setting, and to the best of our knowledge, this is the first result that captures the sample-space trade-off for 'trapping' arms in multi-armed bandits, and it can be of independent interest.

## 1   Introduction

The stochastic multi-armed bandits (MABs) is one of the most well-known models in machine learning and theoretical computer science. In this model, we are given $n$ arms with unknown reward

---

[*] Author names are in alphabetical order.

36th Conference on Neural Information Processing Systems (NeurIPS 2022).

distributions, and we are required to find some arms for certain goals. The *pure exploration* version of the stochastic MABs studies how to find the best arm(s), and it has been extensively investigated in the literature [20, 33, 6, 25, 26, 23, 13, 28, 11, 1, 15]. The model captures many fundamental problems in finding the best items among many, has various applications in experiment design [36, 9], crowdsourcing [16, 43, 10], search ranking [19, 3, 35], networking [42, 40], and others.

We study the problem of finding the best (*one*) arm in stochastic MABs. Since the distribution of the rewards is unknown, the natural strategy is to pull each arm multiple times and use the empirical rewards to find the best arm. Accordingly, the goal of the algorithms in this problem is to identify the best arm with a high constant probability and a number of arm pulls (sample complexity) that is as small as possible. Following the classical upper bound of [20] and lower bound of [33], it is well known that $\Theta(n/\Delta_{[2]}^2)$ arm pulls are both necessary and sufficient – throughout, $\Delta_{[i]}$ is the gap between the rewards of the best and the $i$-th-best arms. Going beyond worst-case, the main focus of the literature has been on obtaining *instance-sensitive* algorithms for the problem, resulting in algorithms of [23] and [26] with $O(\sum_{i=2}^{n} \frac{1}{\Delta_{[i]}^2} \log\log\left(\frac{1}{\Delta_{[i]}}\right))$ arm pulls (see also [15] and references therein for various extensions). This upper bound has at most a doubly-logarithm gap from known instance-optimal lower bounds [33, 23, 15].

Almost the entirety of algorithms in the literature for best arm identification require the entire set of the arms to be repeatedly visited, and therefore, a memory of $\Omega(n)$ arms is inevitably needed. More recently, motivated by applications to processing massive number of arms (e.g., in online advertisements) or the situations when the arms are arriving online (e.g. crowd-sourcing), the *space* requirements of algorithms for stochastic MABs has also received considerable attention [5, 31, 32, 12, 30, 2]. In particular, [5] recently formulated and studied the problem of pure exploration in *streaming* stochastic multi-armed bandits. In their model, the arms arrive one by one in a stream, and the algorithm can only output or pull a past arm if it is stored. The largest number of arms stored at any point is named the *space complexity*; see also [31, 30, 24]. [5] designed an algorithm that given the knowledge of $\Delta_{[2]}$, finds the best arm with high constant probability using $O(n/\Delta_{[2]}^2)$ arm pulls and a memory of a *single* arm.

The algorithm of [5] achieves the worst-case optimal sample and space complexity simultaneously, thus asserting that no trade-off is needed between the sample and space for this problem. On the flip side, this upper bound is only worst-case optimal and *not* instance-sensitive. In a recent result, [24] shows that if we are allowed to make $O(\log(1/\Delta_{[2]}))$ passes over the stream, it is possible to find the best arm with the instance-sensitive sample complexity in the same spirit of [23, 26], and only use a single-arm memory. However, allowing $O(\log(1/\Delta_{[2]}))$ passes seems to be a strong condition especially in streaming applications. Therefore, a natural open question is *whether we can achieve instance-sensitive sample complexity, while maintaining a small memory, preferably still a single arm, in a single pass of the streaming model.*

The main result of this paper is that finding the best arm with the instance-sensitive sample complexity in a single pass is indeed *hard*: if we are only given the parameter $\Delta_{[2]}$, any single-pass streaming algorithm must either use asymptotically higher sample complexity than the instance-sensitive bound, or use a memory of $\Omega(n)$ arms. Furthermore, the lower bound holds even if we are given additional strong assumption of either ($i$) the arms arrive in a random order or ($ii$) the parameters of all reward gaps $\{\Delta_{[i]}\}_{i=2}^n$. En route to proving the above lower bounds, we also provide an *even stronger* lower bound for single-pass streaming algorithms without any parameter (not even $\Delta_{[2]}$): we show that in such a case, any single-pass streaming algorithm with a memory of $o(n)$ has to use an *unbounded* sample complexity. Finally, we complement our lower bounds with a single-pass streaming algorithm that uses a single-arm memory and achieves the instance-sensitive sample complexity, if given the parameter $\Delta_{[2]}$ and conditions of *both* random order stream and a known quantity of the instance-sensitive sample complexity, where the latter is a weaker condition than ($ii$).

## 1.1 Instance-sensitive Sample Complexity

As stated earlier, the worst-case sample complexity of stochastic MABs is $\Theta(n/\Delta_{[2]}^2)$. However, in most instances of interest, one can hope for much better sample complexity that exploits the rewards of different arms in a more fine-grained way. This is the basic idea behind *instance-sensitive* algorithms that have received significant attention in the context of stochastic MABs. We follow the same approach in this paper. In particular, we aim to achieve the sample complexity

of $O\big(\sum_{i=2}^{n} \frac{1}{\Delta_{[i]}^2} \log\log(\frac{1}{\Delta_{[i]}})\big)$ arm pulls – the same bound achieved by classical algorithms for non-streaming stochastic MABs [23, 26].

Following the standard notation used in the literature [23, 26, 13], we denote the reward gap between the best and the $i$-th best arm as $\Delta_{[i]}$. As such, we use $\{\Delta_{[i]}\}_{i=2}^{n}$ to denote the set of *all* gap parameters. The standard instance-sensitive complexity is as follows:

$$\mathbf{H}_1 := \sum_{i=2}^{n} \frac{1}{\Delta_{[i]}^2}, \qquad \mathbf{H}_2 := \sum_{i=2}^{n} \frac{1}{\Delta_{[i]}^2} \log\log\left(\frac{1}{\Delta_{[i]}}\right).$$

The algorithms of [23, 26] use $O(\mathbf{H}_2)$ arm pulls. It is worth noting that no known algorithm has achieved the $O(\mathbf{H}_1)$ sample complexity, and whether $\omega(\mathbf{H}_1)$ sample complexity is necessary is an open problem even in the offline setting. As such, we only focus on the $O(\mathbf{H}_2)$ sample complexity in this paper. The term '(almost) instance-sensitive' is derived from 'instance-optimal' in the literature [13, 1, 5]. We use such a term as an analogy to the 'truly instance-optimal' $O(\mathbf{H}_1)$ sample complexity, and this term is shared by recent work like [27]. Moreover, since there is a constant-factor difference between $\mathbf{H}_2$ and the instance-sensitive sample complexity (i.e., $\mathbf{H}_2$ vs. $O(\mathbf{H}_2)$), we denote the latter as *INST-complexity* in this work.

### 1.2 Our Contributions

Our main contribution is a set of lower bounds showing that it is almost impossible to achieve the instance-sensitive sample complexity with low memory for any single-pass streaming algorithm. Moreover, if no parameter is provided (especially $\Delta_{[2]}$ is not known), it is impossible to achieve any bounded sample complexity with low memory:

> **Result 1** (**Informal**). *There are families of stochastic multi-armed bandits instances such that:*
>
> $(i)$ *If the knowledge of $\Delta_{[2]}$ is provided, but the knowledge of* all other *gap parameters $\{\Delta_{[i]}\}_{i=3}^{n}$ and the random arrival of arms are **not both** provided, the sample complexity of any single-pass streaming algorithm with $o(n)$-arm memory is $\omega(\mathbf{H}_2)$.*
>
> $(ii)$ *If the knowledge of $\Delta_{[2]}$ is not provided and no other parameter is provided, the sample complexity of any single-pass streaming algorithm with $o(n)$-arm memory is **unbounded** as a function of $n$ and $\Delta_{[2]}$;*

Note that $(i)$ of Result 1 captures three cases: only $\Delta_{[2]}$ is known; $\Delta_{[2]}$ is known and the stream is in a random order; and all the gap parameters $\{\Delta_{[i]}\}_{i=2}^{n}$ are known (but the stream is in an adversarial order). Compared to the positive results in [5], our lower bounds rule out any instance-sensitive algorithms under the same parameter settings. Moreover, they provide a strong justification for the knowledge of $\Delta_{[2]}$ in the algorithm of [5] (and any streaming MAB algorithms).

Another remark of Result 1 is the *separation* of the complexity for multi-armed bandits under the streaming and the classical models. The algorithm in [5] implies that with the worst-case sample complexity and a known $\Delta_{[2]}$, there is no separation of hardness for pure exploration MABs between the streaming and the classical settings, i.e. no sample-space trade-off. On the other hand, our results suggest that if we switch to the regime of instance-sensitive sample complexity, or if we remove the assumption of $\Delta_{[2]}$, a sample-space trade-off starts to emerge (note that there exist algorithms that achieve $O(\mathbf{H}_2)$ sample complexity *without* the knowledge of $\Delta_{[2]}$ under the classical setting).

To complement the lower bounds, we design an algorithm that uses a single arm memory and achieves instance-sensitive sample complexity when $\Delta_{[2]}$ is not too small. The algorithm is conditioned on the knowledge of $\Delta_{[2]}$ and *both* the random order stream and the knowledge of INST-complexity. The latter condition, combined with the knowledge of $\Delta_{[2]}$, is a weaker form of all gap parameters – one can certainly compute $\mathbf{H}_2$ if parameters $\{\Delta_{[i]}\}_{i=2}^{n}$ are given.

> **Result 2** (**Informal**). *Assuming the knowledge of $\Delta_{[2]}$ and INST-complexity, and when the arms arrive in a random order, there is a streaming algorithm that finds the best arm in streaming stochastic multi-armed bandits with $O\left(\mathbf{H}_2 + \mathrm{poly}(\frac{\log n}{\Delta_{[2]}})\right)$ arm pulls and a memory of a single arm with high (constant) probability.*

We should note that the sample complexity of our algorithm in Result 2 has an additive term compared to the $O(\mathbf{H}_2)$ obtained in the non-streaming setting [23, 26]. This term however is typically negligible

except for pathological cases where $\Delta_{[2]}$ is extremely small compared to both average values of $\Delta_{[i]}$'s and the number of arms.

A key ingredient of our proof towards Result 1 is an arm-trapping lemma that characterizes the hardness to *keep* the arm in the memory, shown as below:

> **Result 3** (**Arm-trapping lemma, informal**). *There are families of stochastic multi-armed bandits instances such that any algorithm that* outputs $o(n)$ *arms that* contains the index *of the best arm (with high constant probability) has to use* $\Omega\left(\frac{n}{\Delta_{[2]}^2}\right)$ *arm pulls. As a result, any single-pass streaming algorithm that keeps the best arm by the end of the stream (with high constant probability) has to use either* $\Omega\left(\frac{n}{\Delta_{[2]}^2}\right)$ *arm pulls or an* $\Omega(n)$*-arm memory.*

To the best of our knowledge, this is the first Multi-armed Bandits lower bound that explicitly establishes trade-off between sample and space complexity for *trapping* an arm[2]. Note that this statement is stronger than the usual lower bounds on *finding* arms: our lower bound holds even if the algorithm is only required to output a collection of indices that contains the index of the best arm (possibly without *knowing* which arm is the best).

### 1.3 Related Work

Originated in the work of Robbins [36] almost 70 years ago, the study of stochastic MABs basically follows two popular branches: the pure exploration problem, which aims to find the best arm [20, 33, 13, 28, 11, 1, 15]; and the exploration-exploitation problem, which focus on the regret minimization [41, 8, 29, 12, 18]. Covering the vast literature on regret minimization is beyond the scope of this paper but we should note that even regret minimization is studied in streaming setting although mostly for *multi-pass* algorithms [31, 12, 30, 2].

The worst-case optimal bounds on the sample complexity of exploration problem was established by [20] (upper bound) and [33] (lower bound). Besides the best arm identification, [20] also introduced the $\varepsilon$-best arm variation of the pure exploration problem. The problem is motivated by learning under the PAC$(\varepsilon, \delta)$-framework, where the algorithm aims to find an arm with reward at most $\varepsilon$-far from the best arm. The $\varepsilon$-best arm problem is also studied extensively. It has recently been considered in the streaming model as well by [5, 31, 24] who designed streaming algorithms for $\varepsilon$-best arm problem with $O(1)$-arm memory and $O(n/\varepsilon^2)$ arm pulls[3] (we note that [31] also designed a heuristic for this problem on random streams with $O(1)$ memory that works well empirically but does not have the same theoretical guarantee).

The instance-sensitive sample complexity of $O\left(\mathbf{H}_2\right)$ was first achieved by [26], followed up by [23] who achieved the same asymptotic complexity but a better performance in practice. [23] also discovered that for the problem with 2 arms, $\Omega(\frac{1}{\Delta_{[2]}^2} \log\log(\frac{1}{\Delta_{[2]}}))$ arm pulls are necessary – a lower bound that is due to a classical result of [21] from over half of a century ago. [23] conjectured that the lower bound holds for $n$ arms (and the complexity will be 'instance-*optimal*'); nonetheless, a later work by [13] shows that for some instances, an algorithm with $O(\sum_{i=2}^n \frac{1}{\Delta_{[i]}^2})$ arm pulls is possible and formulated a 'gap-entropy' conjecture for capturing the instance-optimal bounds (see [15] for further progress on this conjecture).

## 2 Preliminaries

**Problem Definition**

The formal description of the stochastic multi-armed bandit model that we study is as follows.

**Problem 1** (Pure exploration MABs). *There is a set of $n$ arms $\{\mathsf{arm}_i\}_{i=1}^n$ with unknown rewards, each following a sub-Gaussian distribution with means $\{p_i\}_{i=1}^n$. Our goal is to find the best arm with the highest expected reward, i.e., $p_i$, by pulling them and observing the empirical rewards.*

By definition, we need to solve Problem 1 with randomized algorithms. As such, we aim to design algorithms that find the best arm with *high constant probability*. The reason to not consider the high

---

[2]A concurrent work [2] presents a streaming MABs lower bound with a different sample-space trade-off.

[3]The algorithms in [5, 31] uses $\log^*(n)$-arm memory. A newer version of the algorithm in [5] achieves $O(1)$-arm memory, and [24] takes a different approach which arrives at the same asymptotic guarantees.

probability of $1 - \frac{1}{\text{poly}(n)}$ is that it is impossible to get such a high success probability without an $O(\log(n))$ multiplicative factor to the sample complexity, which is considered large in the literature.

We study Problem 1 under the *streaming* model, which means the arms arrive one by one from the set; for each arriving arm, the algorithm can pull it multiple times to observe the empirical reward or store it to the memory (so it can be pulled later). However, if an arm is not stored, it is lost forever and the algorithm can neither pull nor restore it. We refer to the number of arm pulls by the algorithm as the *sample complexity*, and the maximum number of arms stored as the *space complexity*.

The sample complexity of stochastic MABs is characterized by the *gap parameters* $\{\Delta_{[i]}\}_{i=2}^n$, which denotes the difference between the reward of the best arm and the arm with the $i$-th highest reward. By definition, we have $(\Delta_{[2]} \leq \Delta_{[3]} \leq \cdots \leq \Delta_{[n]})$. We assume that these parameters are greater than 0 – an assumption that is common in the literature [20, 25, 17, 38]. Note that the gap parameters are not necessarily *given* to the algorithm, although it is common to assume the knowledge of $\Delta_{[2]}$. In fact, we provide a comprehensive treatment to the necessity of the $\Delta_{[2]}$ in this work.

It is impossible to cover all of the wide range of assumptions used in the MABs literature. As such, we restrict our attention to the following set of assumptions and conditions.

$(i)$ The knowledge of $\Delta_{[2]}$.

$(ii)$ The (uniform) random order of arm arrival.

$(iii)$ The knowledge of additional gap parameters $\{\Delta_{[i]}\}_{i=3}^n$.

$(iv)$ The knowledge of the INST-complexity $\mathbf{H}_2$ (or $\mathbf{H}_2^\delta$): the instance-sensitive sample complexity.

Note that conditions $(i)$ and $(iii)$ forms a stronger condition than conditions $(i)$ and $(iv)$ – the latter can be computed given the former, but *not* vice versa.

## 3 The Arm Trapping Lemma

Our lower bounds are built upon an arm trapping lemma, which is a information-theoretic sample-space trade-off for any algorithm to 'trap' the best arm (*not* restricted to the streaming setting). Compared to the classical lower bounds for finding arms in stochastic MABs, our lemma stands out by showing that even *trapping* the best arm (possibly without knowing which one is the best among the trapped arms) requires a lot of arm pulls. As a direct corollary, we can show that for any streaming algorithm with a memory of $o(n)$ arms, *storing* the best arm in the stream takes $\Omega(\frac{n}{\Delta_{[2]}^2})$ arm pulls.

Our hard distribution for the arm-trapping lemma is quite natural: consider a collection of $n$ arms with one of them having a reward of $\frac{1}{2}$, and all other arms having a reward of $\frac{1}{2} - \beta$ for some $\beta$. The high level idea behind the proof is a non-standard reduction from this problem to the problem of distinguishing whether the reward of a single arm is $1/2$ or $1/2 - \beta$; a problem that provably requires $\Omega(1/\beta^2)$ samples. The reduction places this 'special arm' in a collection of $n - 1$ arms with reward $1/2 - \beta$, uniformly permute them, and then gives them as input to any algorithm that can supposedly contradict Lemma 3.1. The proof then shows that if the algorithm can only place this special arm with non-trivial probability among the output 'trap' set, this arm actually has reward of $1/2$ and not $1/2 - \beta$. Combining this with a simple 'direct sum style' argument then proves the lower bound.

We now formally define the hard family of instances as follows.

---

`DIST`: **A hard distribution for trapping the best arm**

    1. An index $i^\star$ sampled uniform at random from $[n]$.

    2. For $i \neq i^\star$, let the arms be with reward $p_i = \frac{1}{2} - \beta$.

    3. For $i = i^\star$, let the arm be with reward $p_{i^\star} = \frac{1}{2}$.

---

Based on the definition of `DIST`, we can give the formal statement of our streaming arm trapping lemma.

**Lemma 3.1** (Arm Trapping Lemma)**.** *Any algorithm that outputs (indices of) $\frac{n}{8}$ arms which contains the best arm on* `DIST` *with probability at least $\frac{2}{3}$ has to use at least $\frac{1}{1200} \cdot \frac{n}{\beta^2}$ arm pulls.*

The proof of Lemma 3.1 follows a (non-standard) reduction from the complexity to distinguish two arms with a $\beta$-reward gap, specified as follows:

**Lemma 3.2.** *Consider an arm with the reward from the following distribution*

- $p = \frac{1}{2}$, *with probability* $\frac{1}{2}$;
- $p = \frac{1}{2} - \beta$, *with probability* $\frac{1}{2}$.

*Any algorithm to determine the reward of the arm with $\beta < \frac{1}{6}$ and a success probability of at least $\frac{7}{12}$ has to use $\frac{1}{144} \cdot \frac{1}{\beta^2}$ arm pulls.*

Lemma 3.2 is an information-theoretic lower bound similar to the ones in [33]. The bound shows that for any algorithm to distinguish an arm with reward either $\frac{1}{2}$ or $(\frac{1}{2} - \beta)$, it has to pull the arm $\frac{n}{\beta^2}$ times (if $\beta$ is sufficiently small). This argument is standard and is provided for completeness as similar results, such as classical Farrell's bound [21, 23, 13], have just subtle differences with this bound, which in fact prohibits us from using them. Limited by space, we defer the proofs of Lemma 3.1 and Lemma 3.2 to Appendix C.

As a natural corollary of Lemma 3.1, any single-pass streaming algorithm to *store* the best arm with a memory of $o(n)$ arms has to use $\Omega\left(\frac{n}{\beta^2}\right)$ arm pulls.

**Corollary 3.3.** *Any streaming algorithm that stores the best arm on* DIST *with a memory of $\frac{n}{8}$ arms with probability at least $\frac{2}{3}$ has to use $\Omega\left(\frac{n}{\beta^2}\right)$ arm pulls.*

Corollary 3.3 gives a powerful tool to prove space lower bounds in streaming MABs by exploiting the dilemma between necessity of the exploration of best arm in the early part of the stream and the cost of doing so. In particular, consider instances where $O(n)$ arms follow DIST in Corollary 3.3 appear in the first part of the stream, and a few arms by the end of the stream are either with very high or with very low rewards. If the algorithm explores the best arm in the earlier stream, it risks to 'waste' a large number of arm pulls since the late arm high-reward case renders the exploration unnecessary. On the other hand, if the algorithm decides to forgo the best arm in the earlier part, it runs the risk that the late arms are with very low rewards, and the best arm will be missed.

# 4 Streaming Lower Bounds for Multi-armed Bandits with Instance-sensitive Sample Complexity

In this section, we provide the main lower bounds for single-pass streaming MABs. On the high level, our lower bound states that for any streaming algorithm with a memory of $o(n)$ arms that finds the best arm, if some parameters are not given, the sample complexity of the algorithm can be far more than the instance-sensitive $\Theta(\mathbf{H}_2)$ or even unbounded.

**Remark 4.1.** To be consistent with the definition of streaming MAB, in our lower bounds, we only measure the space of the algorithms in terms of the number of arms they store; the algorithms otherwise are allowed to maintain any arbitrary statistics on the input. What we show in all these lower bounds is that if the sample complexity is "small", then at some point the algorithm must have mistakenly discarded the best arm from its memory without the knowledge of even the index of this arm; thus at the end of the stream, the algorithm has no way of recovering the best arm (even given the arbitrary statistics it has collected).

## 4.1 A lower bound without the knowledge of $\Delta_{[2]}$ and other parameters

We begin with showing that if no additional parameter is provided other than the stream itself, the sample complexity can be *arbitrarily* high, and even $O(n/\Delta_{[2]}^2)$ arm pulls are not sufficient. In fact, we show that in this case, the sample complexity cannot be bounded as a function of $n$ and $\Delta_{[2]}$. In contrast, the algorithm in [5], which assumes the knowledge of $\Delta_{[2]}$, achieves $O(n/\Delta_{[2]}^2)$ arm pulls with the memory of a single arm. Therefore, our lower bound asserts the importance of $\Delta_{[2]}$ for exploring the best arm under the streaming model.

**Theorem 1.** *For any function $f : (0, 1) \to (0, 1)$, there always exists a family of streaming stochastic multi-armed bandit instances, such that any algorithm to find the best arm with a memory of $\frac{n-1}{8}$ arms and a success probability of at least $\frac{5}{6}$ has to use $\omega\left(\frac{n}{f(\Delta_{[2]})}\right)$ arm pulls, if no further knowledge other than the arms themselves are provided.*

On the high level, the adversarial distribution for Theorem 1 is as follows. We let the first $(n - 1)$ arms in the stream to follow DIST in Corollary 3.3, and the last arm to have either a very high or a

very low reward, with probability $\frac{1}{2}$ each. As such, to guarantee a high enough success probability, the algorithm needs to trap the arm with the $p = \frac{1}{2}$ reward among the first $(n-1)$ arms. However, the sample complexity to keep such an arm becomes unbounded when the last arm is with a very high reward. We defer the proof of Theorem 1 to Appendix B.1.

## 4.2 Instance-sensitive Lower bounds with the knowledge of $\Delta_{[2]}$

Going beyond the case where none of the conditions are given, we now show the lower bound with $\Delta_{[2]}$ provided. Recall that [5] already solved the problem with $O(\frac{n}{\Delta_{[2]}^2})$ arm pulls; therefore, it is pointless to expect any result as strong as Theorem 1. Instead, we prove lower bound of $\omega(\mathbf{H}_2)$ by exploiting the gap between $\Theta\left(n/\Delta_{[2]}^2\right)$ and $\Theta\left(\mathbf{H}_2\right)$. We first present the lower bound with *only* the knowledge of $\Delta_{[2]}$.

**Theorem 2.** *There exists a family of streaming stochastic multi-armed bandit instances, such that given the knowledge of $\Delta_{[2]}$, any algorithm to find the best arm with a memory of $\frac{n-2}{8}$ arms and a success probability of at least $\frac{5}{6}$ has to use $\omega\left(\mathbf{H}_2\right) = \omega\left(\sum_{i=2}^n \frac{1}{\Delta_{[i]}^2} \log\log\left(\frac{1}{\Delta_{[i]}}\right)\right)$ arm pulls.*

We have noted that it is pointless to expect any result as strong as Theorem 1. Indeed, the hard distribution in Section 4.1 does not work anymore – by the change of the reward of the final arm, the value of $\Delta_{[2]}$ is $\beta$ in one case and $\frac{1}{6} >> \beta$ in the other. Therefore, by simply checking the parameter, an algorithm can determine whether to store the best arm among the first $(n-1)$ arms, which makes the distribution impossible to 'fool' the algorithm.

As an alternative idea, we modify the distribution of the instances to satisfy the following properties. Firstly, it forces the algorithm to make $\Omega(\frac{n}{\Delta_{[2]}^2})$ arm pulls to find the best arm. Secondly, it leaves a large gap between $\Delta_{[2]}$ and other $\Delta_{[i]}$, $\forall i \geq 2$ such that there is $\frac{n}{\Delta_{[2]}^2} = \omega(\mathbf{H}_2)$. And finally, $\Delta_{[2]}$ is static (the same version) for all the instances.

Limited by space, we defer the formal proof of Theorem 2 to Appendix B.2. We also remark that the proof of Theorem 2 works under an *approximate knowledge* of $\Delta_{[2]}$, which can lead to slightly stronger lower bounds (see Remark B.2 for details).

We now provide lower bounds with either the Random Arrival or the quantity of $\mathbf{H}_2$ (but *not both*) in Theorem 3 and Theorem 4.

**Theorem 3.** *There exists a family of streaming stochastic multi-armed bandit instances, such that given the knowledge of $\Delta_{[2]}$ and the random arrival of arms, any algorithm to find the best arm with a memory of $(\frac{n}{24} - 1)$ arms and a success probability of at least $\frac{80}{81}$ has to use $\omega\left(\mathbf{H}_2\right) = \omega\left(\sum_{i=2}^n \frac{1}{\Delta_{[i]}^2} \log\log\left(\frac{1}{\Delta_{[i]}}\right)\right)$ arm pulls.*

**Theorem 4.** *There exists a family of streaming stochastic multi-armed bandit instances, such that given the knowledge of all gap parameters $\{\Delta_{[i]}\}_{i=2}^n$, any algorithm to find the best arm with a memory of $\frac{t}{8}$ arms for $t = o(n)$ and a success probability of at least $\frac{8}{9}$ has to use $\Omega\left(\frac{n}{t} \cdot \mathbf{H}_2\right) = \omega\left(\mathbf{H}_2\right)$ arm pulls.*

The idea to prove the lower bound with random arrival is quite simple: we show that with at least a constant probability, the distribution of an instance with random arrival will resemble the instance we used for Theorem 2. More specifically, suppose each arm arrives at any position with probability $\frac{1}{n}$. Then, with probability $\frac{1}{3}$, the arm with reward $\frac{1}{2}$ appears in the first $\frac{1}{3}$-fraction of the stream; Similarly, with probability $\frac{2}{3} \cdot \frac{2}{3}$, both of the last two arms in the distribution of Theorem 2 will not pop up until the late $\frac{2}{3}$ fraction of the stream. The family of the instances with both of the above properties indeed resembles the distribution we used earlier, and we can show that the algorithm is forced to use a lot of arm pulls under this scenario. The hard instance for Theorem 4 is developed with a different idea. Due to space limits, we defer the proofs to Appendix B.

## 5 An Instance-sensitive Streaming Algorithm

We begin with introducing notations used in this section. We denote the best arm with $\mathsf{arm}^*$. Furthermore, we denote the instance-sensitive complexity (*INST-complexity*) for solving the problem

with probability of success at least $1 - \delta$ as $\mathbf{H}_2^{\delta} := \sum_{i=2}^{n} \frac{1}{\Delta_{[i]}^2} \log\left(\frac{1}{\delta} \log\left(\frac{1}{\Delta_{[i]}}\right)\right)$. Note $\mathbf{H}_2$ and $\mathbf{H}_2^{\delta}$ only differ from the correct probability; if one sets $\delta$ to be a constant, $\mathbf{H}_2$ and $\mathbf{H}_2^{\delta}$ become the same asymptotically.

As mentioned before, our algorithm makes two assumptions on the stream of arms, namely:

($i$) *Random arrival stream*: the stream is a random permutation of the arms $\{\mathsf{arm}_i\}_{i=1}^{n}$.

($ii$) *Known INST-complexity*: The parameter $\mathbf{H}_2^{\delta}$ (or an $\alpha$-approximation of it with $\alpha = O(1)$ being a *fixed and known* constant) is given;

Assumption ($ii$) is a weaker version of all gap parameters $\{\Delta_{[i]}\}_{i=2}^{n}$ used in the lower bound of Theorem 4. If the gap parameters are given, we can simply compute $\mathbf{H}_2$ and $\mathbf{H}_2^{\delta}$. The reverse direction is not generally possible.

Although the assumptions are strong, they are common in existing work of streaming models and stochastic MABs. The random order stream is a standard assumption in the streaming setting (see [7, 4, 34, 39], to name a few). The knowledge of $\mathbf{H}_2^{\delta}$ is similar-in-spirit (albeit not identical) to the *fixed-budget* stochastic MABs, which has been studied intensively in the literature; see, e.g. [26, 11]. Furthermore, from a practical point of view, the knowledge of $\Delta_{[2]}$ can be satisfied by an estimation that is at most $\Delta_{[2]}$. The stronger assumptions of ($i$) and ($ii$) can be satisfied under the following scenarios:

- The arms are stored in a 'bank' and a local machine reads the arms with a small working memory. This allows the arms to be 'served' in a random order from the external bank.

- There exists historical data such that one can roughly estimate the quantity of INST-complexity.

While the scenarios are restricted, it does capture a non-negligible range of applications. The formal guarantees of our algorithm is as follows.

**Theorem 5.** *There is a streaming algorithm that for any confidence parameter $\delta > 0$, given $n$ arms in a random order, gap parameter $\Delta_{[2]}$, and INST-complexity $\mathbf{H}_2^{\delta} = \sum_{i=2}^{n} \frac{1}{\Delta_{[i]}^2} \log\left(\frac{1}{\delta} \log\left(\frac{1}{\Delta_{[i]}}\right)\right)$, finds the best arm w.p. at least $1 - \delta$ using $O\left(\mathbf{H}_2^{\delta} + \frac{1}{\Delta_{[2]}^7} \cdot \log(\frac{n}{\delta})^2 \cdot \log^2\left(\frac{1}{\delta} \log\left(\frac{1}{\Delta_{[2]}}\right)\right)\right)$ arm pulls and a memory of a single arm.*

The algorithm in Theorem 5 contains an additive term in sample complexity beside the standard $\mathbf{H}_2^{\delta}$ term. For most reasonable choices of parameters, this additive term is in fact negligible — in other words, our streaming algorithm in Theorem 5 achieves the same sample complexity as that of non-streaming algorithms of [26, 23]. From a practical perspective, assuming $\mathbf{H}_2 \geq 100n$ and a constant success probability, we only need $n$ to be in $10^6 \sim 10^7$ and $\Delta_{[2]}$ be around $0.1 \sim 0.2$ to make $\mathbf{H}_2$ the dominating term, which seems reasonable in practice.

### 5.1 The Algorithm

Since we only maintain a single arm memory through the stream, we define the stored arm as the king in a manner similar to the GAME-OF-COINS algorithm of [5] (an arm with Bernoulli reward is equivalent to a *coin*). We also adopt the *challenge procedure* first developed by [5]: for each arriving arm, if there already exists a king, the new arm will begin a challenge procedure. If the challenge is successful, the challenger becomes the new king to be challenged by the later arms; otherwise, the challenger is discarded, and the existing king retains its position and will be challenged by later arms. By the end of the stream, the algorithm outputs the king that is finally stored.

The crucial part of our algorithm (similar to GAME-OF-COINS of [5]) is how to design the challenge process. Following [5], the generic strategy for this is to compare the empirical reward on many *levels*. That is, on each level, we pull both the king and the challenger for a certain times, and observe which arm has a lower empirical reward, to identify the *defeated* arm. If the challenger is defeated, we can simply discard it; but, if the king is defeated, we increase the number of arm pulls and compare the empirical rewards again. The king will not be discarded until some certain conditions are met.

The challenge procedure in [5] inherently requires $\Omega(1/\Delta_{[2]}^2)$ arm pulls for each challenger even on level 1. Therefore, the algorithm inevitably requires $\Theta(n/\Delta_{[2]}^2)$ arm pulls. Our main task here is to design a new challenge with fewer arm pulls, while maintaining the correctness of the algorithm.

To this end, we now formally introduce the algorithm. Let us first set up the parameters as follows.

$$\left\{\tilde{\Delta}_\ell\right\}_{\ell=1}^\infty : \tilde{\Delta}_\ell = \frac{1}{4} \cdot \left(\frac{1}{2}\right)^{\ell-1} \qquad\qquad \text{(the guess of the gap at the current level)}$$

$$\{\delta_\ell\}_{\ell=1}^\infty : \delta_\ell := \frac{\delta}{50\ell^3} \qquad\qquad \text{(the confidence parameter)}$$

$$\{s_\ell\}_{\ell=1}^\infty : s_\ell := \frac{4}{\tilde{\Delta}_\ell^2} \cdot \log\left(\frac{1}{\delta_\ell}\right) \qquad\qquad \text{(number of pulls)}$$

$$m_{\text{early}} := \frac{30}{\Delta_{[2]}^4} \cdot \log^2\left(\frac{n}{\delta}\right) \cdot \log^2\left(\frac{1}{\delta} \log\left(\frac{1}{\Delta_{[2]}}\right)\right) \qquad \text{(number of the 'warm-up' arms)}$$

$$b := 1800 \cdot \frac{\mathbf{H}_2^\delta}{n}. \qquad\qquad \text{(budget allocated to king for each arriving arm)}$$

We also need some new definitions. We refer to the period of time between the introduction of a new king and the time it is discarded (or the end of the stream) as an *epoch*. For each epoch, we define the *early arms* as the first $m_{\text{early}}$ arms, and the the *late arms* as the arms afterwards. During each epoch, we maintain a fixed estimate for the value of $p^*$ as a parameter $p_{\text{est}}$ that will only be updated after this epoch ends (and $p_{\text{est}}$ is increasing throughout the updates of the algorithm).

We can now present the algorithm:

---

**Main Algorithm:** INSTANCE-SENSITIVE-GAME-OF-ARMS:

    a) Initialize the first arm as king and the first guess of $p^*$ as $p_{\text{est}} = \frac{\Delta_{[2]}}{20}$.

    b) For each epoch:

        $(i)$ Run GAME-OF-COINS of [5] on early arms with parameters:

$$n' := m_{\text{early}} \quad \Delta'_{[2]} = \frac{\Delta_{[2]}}{4} \quad \text{and} \quad \delta' = \frac{\delta}{8}$$

        while setting the reward of the king to be $p_{\text{est}}$ (start their algorithm with an "artificial" first arm/coin with reward $p_{\text{est}}$). Terminate if GAME-OF-COINS discards king and jump to Line $(iii)$; otherwise, initialize the budget of the king as $B \leftarrow b \cdot m_{\text{early}}$.

        $(ii)$ For each late arm, give the king a budget of $b$, i.e., $B \leftarrow B + b$; run **Instance-Sensitive Challenge Subroutine** (below) for king, challenger, budget $B$, and $p_{\text{est}}$.

        $(iii)$ If the king is discarded, set the challenger as king and update $p_{\text{est}}$ as follows:

            (1) Pull the challenger $\frac{64}{\Delta_{[2]}^2} \cdot \log(\frac{n}{\delta})$ times and record the new $\widehat{p}_{\text{challenger}}$.

            (2) Set $p_{\text{est}} \leftarrow \max(\widehat{p}_{\text{challenger}}, p_{\text{est}} + \frac{\Delta_{[2]}}{20})$;

    c) Run till the end of the stream, and output the king as the best arm.

---

**Instance-Sensitive Challenge Subroutine**:

**Input:** the king and its budget $B$; a challenger arm; the estimation parameter $p_{\text{est}}$.

    1. For levels $\ell = 1, 2, \cdots$, as long as $s_\ell < \frac{64}{\Delta_{[2]}^2} \cdot \log(\frac{n}{\delta})$:

        (a) If $B > s_\ell$, let $B \leftarrow B - s_\ell$ and pull *challenger* $s_\ell$ times to get empirical reward $\widehat{p}$.

        (b) If $\widehat{p} < p_{\text{est}} - \Delta_\ell/4$, drop the challenger and terminate.

    2. Pull the challenger $(\frac{64}{\Delta_{[2]}^2} \cdot \log(\frac{n}{\delta})$ times and record its empirical reward $\widehat{p}$:

        (a) If $\widehat{p} > p_{\text{est}} - \Delta_{[2]}/4$, declare the challenge successful, and discard the king; otherwise, declare the challenge unsuccessful, and discard the challenger.

---

In a nutshell, the intuitions for the algorithm are as follows. Classical offline algorithms for instance-sensitive explorations (e.g. the algorithm in [26]) often use estimations $\tilde{\Delta}_\ell$ for gaps $\Delta_{[i]}$, and argue

that by iteratively comparing the arm rewards with the best arm, the distinction between the best and the $i$-th best arms becomes very clear after $\tilde{\Delta}_\ell \leq \Delta_{[i]}$. The value of $\tilde{\Delta}_\ell$ takes $O(\log \frac{1}{\Delta_{[i]}})$ iterations to converge; as such, it takes $O\big(\frac{1}{\Delta_{[i]}^2} \log \log(\frac{1}{\Delta_{[i]}})\big)$ arm pulls to distinguish the best and the $i$-th best arms. The strategy inherently requires the best arm to be stored in the beginning of the algorithm.

Alas, migrating to the streaming setting, the above strategy no longer works: we cannot assume the best arm is stored in the beginning – it may appear in a very late part of the stream. As such, we face the challenges of $(i)$. it is unclear how to use the reward of the best arm as in the offline setting; and $(ii)$. the comparisons between sub-optimal arms may take too many arm pulls.

It turns out our assumptions are helpful to tackle the above challenges. On the very high level, we use the knowledge of $\Delta_{[2]}$ to 'guess' the reward of the best arm – this allows us to iteratively update the value $p_{\text{est}}$ in the algorithm, and eventually arrive at an estimation that is $\big(c \cdot \Delta_{[2]}\big)$-close to the best reward ($c < 1$). Furthermore, we use the the random order of the stream and the knowledge of $\mathbf{H}_2^\delta$ to 'budget' the number of arm pulls for each arriving arm. It turns out this allows us to argue the budget is sufficient for the real best arm in an amortized manner.

Limited by space, we defer the detailed overview and the formal analysis of the algorithm to Appendix D.

**Remark 5.1.** Our lower bounds in Section 4 shows that if we are given the knowledge of $\Delta_{[2]}$, but not *both* the random order of arrival and the knowledge of $\mathbf{H}_2^\delta$ (in the stronger form of $\{\Delta_{[i]}\}_{i=3}^n$), it is impossible to arrive at the INST-complexity sample complexity. On the other hand, it leaves the possibility for an algorithm to find the best arm with $o(n)$ memory, assuming no knowledge of $\Delta_{[2]}$ but with given parameters $\{\Delta_{[i]}\}_{i=3}^n$ and the random arm arrival. Indeed, it appears the following modification of our algorithm works for such a scenario: sample each arm in the stream with probability $\frac{1}{2}$, and run INSTANCE-SENSITIVE-GAME-OF-ARMS by replacing $\Delta_{[2]}$ with $\Delta_{[3]}$ and the $\mathbf{H}_2^\delta$ with the quantity $\sum_{i=3}^n \frac{1}{\Delta_{[i]}^2} \log \left( \frac{1}{\delta} \log \left( \frac{1}{\Delta_{[i]}} \right) \right)$. By the sampling of arms, with probability $\frac{1}{8}$, the best arm and third-best arm join the stream, while the second-best arm does not. Therefore, we can run $O\left(\log(1/\delta)\right)$ copies of the algorithm, and output the top arm among the $O\left(\log(1/\delta)\right)$ copies by running the offline algorithm in [26] at the end of the stream. Verifying whether the above strategy works will be an interesting immediate next-step for the work.

## 6  Discussion and Conclusion

In this paper, we studied the limitations for streaming MABs algorithm with the instance-sensitive sample complexity in a single pass. We proved a collection of lower bounds showing that achieving the instance-sensitive sample complexity with a small memory in a single-pass stream is *hard*: with the absence of various (subsets of) assumptions, including the knowledge of $\Delta_{[2]}$, the additional gap parameters $\{\Delta_{[i]}\}_{i=3}^n$, and the random arrival of arms, no algorithm can achieve the instance-sensitive bound with $o(n)$-arm memory. Furthermore, as a byproduct of the lower bounds, we establish the crucial role of $\Delta_{[2]}$ in streaming MABs: without such a parameter, any algorithm with a memory even as large as a constant fraction of $n$ cannot achieve any bounded sample complexity. We complemented our lower bounds by an algorithm that achieves the instance-sensitive bound under most realistic parameters and with a memory of a single arm when (almost) all assumptions hold. Our lower and upper bounds demonstrate a sharp dichotomy.

Our results open the door for various future work in this line of research. One open problem posed by our results is whether one can refine the streaming algorithm (or design a new one) that drops the additive term on the sample complexity. Currently, we do *not* rule out the possibility that the additive term is *necessary*. Moreover, from the application side, an interesting problem is to explore the empirical performances of our algorithm, and observe the numerical advantages compared to the algorithm in [5]. And finally, a more challenging open problem is to understand the tight number of passes that are *necessary* and *sufficient* to achieve the instance-sensitive sample complexity (with no or minimum assumptions) under the streaming MABs model.

### Acknowledgments and Disclosure of Funding

Research supported in part by a NSF CAREER Grant CCF-2047061, a gift from Google Research, and a Rutgers Research Council Fulcrum Award.

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
