# A   Technical Preliminaries

We now present some technical results that will be repeatedly used in the rest of the paper. We start with the standard Chernoff-Hoeffding bound

**Proposition A.1.** *Let $X_1, X_2, \cdots, X_m$ be a sequence of independent random variables bounded in the range $[0, 1]$. Define $S_m = \sum_{i=1}^{m} X_i$, there is*

$$\Pr(|S_m - \mathbb{E}[S_m]| \geq t) \leq 2 \cdot \exp\left(-\frac{2t^2}{m}\right).$$

A direct corollary of the Chernoff-Hoeffding bound (see, e.g. [5] for a proof) on the stochastic multi-armed bandits is as follows.

**Proposition A.2.** *Let $\mathsf{arm}_1$ and $\mathsf{arm}_2$ be two different arms with rewards $p_1$ and $p_2$. Suppose $p_1 - p_2 \geq \theta$ and we pull each arm $\frac{K}{\theta^2}$ times to obtain empirical rewards $\widehat{p}_1$ and $\widehat{p}_2$. Then,*

$$\Pr\left(\widehat{p}_1 \leq \widehat{p}_2\right) \leq 2 \cdot \exp\left(-\frac{1}{4} \cdot K\right).$$

We also use the following variation of Chernoff bound for sampling without replacement.

**Proposition A.3** (cf. [22, 37]). *Let $X_1, X_2, \cdots, X_m$ be a sequence of random variables bounded in the range $[a_i, b_i]$, and let them to be dependent in the form of sampling without replacement. Let $S_m = \sum_{i=1}^{m} X_m$, we have*

$$\Pr(S_m - \mathbb{E}[S_m] \geq t) \leq \exp\left(-\frac{2t^2}{\sum_{i=1}^{m}(b_i - a_i)^2}\right).$$

Furthermore, we use the algorithm by [5] as a subroutine of our algorithm. This algorithm, named GAME-OF-COINS, guarantees the following properties

**Proposition A.4** ([5]). *Given $n$ arms and the gap parameter $\Delta_{[2]}$, algorithm GAME-OF-COINS finds the best arm with probability at least $(1 - \delta)$, a sample complexity of $O\left(\frac{n}{\Delta_{[2]}^2} \cdot \log(\frac{1}{\delta})\right)$ and a memory of a single arm. Furthermore, the algorithm GAME-OF-COINS has the following properties*

$(i)$ **Completeness**: *For any subset of arms, if the stored arm has a reward $p$ that is at least $\Delta$ larger than other arms, then algorithm GAME-OF-COINS does not discard the stored arm with probability at least $(1 - \delta)$ and a sample complexity of $O\left(\frac{n}{\Delta^2} \cdot \log(\frac{1}{\delta})\right)$.*

$(ii)$ **Soundness**: *For any subset of arms, if the stored arm has a reward $p$ that is at least $\Delta$ smaller than $\mathsf{arm}^*$, then algorithm GAME-OF-COINS make $\mathsf{arm}^*$ the stored arm with probability at least $(1 - \delta)$ and a sample complexity of $O\left(\frac{n}{\Delta^2} \cdot \log(\frac{1}{\delta})\right)$.*

# B   Missing Proofs of Section 4 (Lower Bounds)

We provide the lower bound proofs for the results in Section 4. We remark that these lower bounds are are different from standard streaming or multi-armed bandits lower bounds, as they establish sample-space trade-offs. In particular, lower bounds for *offline* multi-armed bandits are often information-theoretic and does *not* depend on adversarial instances. In contrast, to capture the sample-space trade-off in the streaming setting, it is necessary to exploit the order of the stream and use adversarial instances.

## B.1   Proof of Theorem 1

By Yao's minimax principle, it suffices to prove the lower bound for deterministic algorithms over a hard distribution of inputs. Fix any function $f : (0, 1) \rightarrow (0, 1)$, we construct the following distribution

---

**A hard distribution for streaming MAB without any further knowledge**

1. Set the value of $\beta$ such that $\frac{1}{\beta^2} = \omega(\frac{1}{f(\frac{1}{6})})$.

2. Set the first $(n - 1)$ arms in the stream with DIST and provide $\beta$ as we specified above.

---

> 3. Set $\mathsf{arm}_n$ with reward $p_n$ from the followings distribution: with probability $\frac{1}{2}$, $p_n = \frac{2}{3}$; with probability $\frac{1}{2}$, $p_n = \frac{1}{3}$.

Observe that if the algorithm wants to solve the problem with probability at least $\frac{5}{6}$, on $\frac{1}{2}$ of the instances, it must find the best arm with probability at least $\frac{2}{3}$ (i.e. $\frac{2}{3}$-fraction of the instances). This is because otherwise, the overall success probability is less than

$$\frac{2}{3} \cdot \frac{1}{2} + \frac{1}{2} \cdot 1 = \frac{5}{6}. \qquad \text{(assuming the other } \frac{1}{2} \text{ of the instances with a success probability of 1)}$$

For half of the instances, we have $p_n = \frac{1}{3}$, which means the last arm has no way to become the best arm (note that $\beta < \frac{1}{6}$). Therefore, to find the best arm on these instances, we must store the best arm among the first $(n-1)$ arms. By Corollary 3.3, storing the best arm over $\frac{2}{3}$ of the instances with the given memory takes at least $\Omega\left(\frac{n}{\beta^2}\right)$ arm pulls.

Now for the other $\frac{1}{2}$ fraction of the instances, $\mathsf{arm}_n$ is certainly the best arm, and the gap parameter $\Delta_{[2]}$ is $\frac{2}{3} - \frac{1}{2} = \frac{1}{6}$. Note that since the algorithm is deterministic, it also has to store the best arm for $\frac{2}{3}$ of the instances of the first $(n-1)$ arms in this case, which costs $\Omega\left(\frac{n}{\beta^2}\right)$ arm pulls. By the way we pick $\beta$, we have $\frac{n-1}{\beta^2} = \omega(\frac{n}{f(\frac{1}{6})})$ for all these instances.

**Remark B.1.** We remark that even with the random arrival of arms, the sample lower bound in Theorem 1 still holds. The proof is almost identical to the case with the known $\Delta_{[2]}$, and the techniques will be clear once we present the proof of Theorem 3. To avoid redundant repetition of the details, we omit the proof for Theorem 1 with random arrival.

## B.2 Proof of Theorem 2

We consider the following distribution of the streaming multi-armed bandits

> **A hard distribution for streaming MAB with known $\Delta_{[2]}$**
>
> 1. Set a value of $\beta$ such that $\beta << \frac{1}{6}$.
> 2. Set the first $(n-2)$ arms in the stream with DIST and provide $\beta$ as we specified above.
> 3. Set $\mathsf{arm}_{n-1}$ and $\mathsf{arm}_n$ with rewards $p_{n-1}$ and $p_n$ from the followings distribution: with probability $\frac{1}{2}$, $p_{n-1} = \frac{2}{3}$, $p_n = \frac{2}{3} + \beta$; with probability $\frac{1}{2}$, $p_{n-1} = \frac{1}{3}$, $p_n = \frac{1}{3}$.

With an argument identical to the proof of Theorem 1, we can see that for the algorithm to solve the problem with probability at least $\frac{5}{6}$, on $\frac{1}{2}$ of the instances, it must find the best arm with probability at least $\frac{2}{3}$. For half of the instances, we have $p_{n-1} = p_n = \frac{1}{3}$; therefore, any algorithm has to pay at least $\Omega\left(\frac{n}{\beta^2}\right)$ arm pulls to store the best arm among the first $(n-2)$ arms over $\frac{2}{3}$ of the instances. Note that in this case, the best arm is the arm with reward $\frac{1}{2}$, thus we have $\Delta_{[2]} = \beta$.

Now for the other $\frac{1}{2}$ fraction of the instances, $\mathsf{arm}_n$ is the best arm, and the gap parameter $\Delta_{[2]}$ is still $\beta$. Therefore, the instance of the first $(n-1)$ arms is identical to the above case. Since the algorithm is deterministic, it takes $\Omega\left(\frac{n}{\beta^2}\right)$ arm pulls. On the other hand, by the distribution of the rewards of arms, the quantity of $\mathbf{H}_2$ is now $\sum_{i=2}^{n} \frac{1}{\Delta_{[i]}^2} \log\log\left(\frac{1}{\Delta_{[i]}}\right) \leq \frac{1}{\beta^2} + \frac{n-1}{(\frac{1}{6})^2} = o(\frac{n}{\beta^2})$. Therefore, on half of the instances, the algorithm has to use $\omega\left(\sum_{i=2}^{n} \frac{1}{\Delta_{[i]}^2} \log\log\left(\frac{1}{\Delta_{[i]}}\right)\right)$ arm pulls. Applying Yao's minimax principle gives us the worst-case sample complexity.

**Remark B.2.** It is not difficult to see that the above proof works for the approximate knowledge of $\Delta_{[2]}$. In particular, if we are given an $\alpha$-approximation for $\Delta_{[2]}$ ($\alpha > 1$), we can change the gap between $p_{n-1}$ and $p_n$ to $\alpha \cdot \beta$. As a result, the $\Omega\left(\frac{n}{\beta^2}\right)$ sample complexity becomes $\Omega\left(\alpha^2 \cdot \frac{n}{\Delta_{[2]}^2}\right)$.

## B.3 Proof of Theorem 3

We consider the following distribution of the streaming multi-armed bandits with the random arrival of arms:

---

**A hard distribution for streaming MAB with known $\Delta_{[2]}$ and random arrival**

1. Set $\beta$ such that $\beta << \frac{1}{6}$, and let $(n-2)$ arms be following DIST with parameter $\beta$.
2. Set 2 arms arm$'$ and arm$''$ with rewards $p'$ and $p''$ from the followings distribution: with probability $\frac{1}{2}$, $p' = \frac{2}{3}, p'' = \frac{2}{3} + \beta$; with probability $\frac{1}{2}$, $p' = \frac{1}{3}, p'' = \frac{1}{3}$.
3. Order the arms by uniformly at random picking a permutation over $[n]$.

---

We first show that with probability $\frac{4}{27}$, the following three events happen together: $(i)$. the arm with reward $\frac{1}{2}$ appears in the first $\frac{1}{3}$ of the stream; $(ii)$. arm$'$ appears in the last $\frac{2}{3}$ of the stream; and $(iii)$. arm$''$ appears in the last $\frac{2}{3}$ of the stream. To see this, note that by picking the permutations uniformly at random, each arm as $\frac{1}{n}$ probability to appear at a position $j$. Therefore, the probability for the above events to happen is at least $\frac{1}{3} \cdot \frac{2}{3} \cdot \frac{2}{3} \geq \frac{4}{27}$.

For the algorithm to solve the problem with probability at least $\frac{26}{27}$, on the above (at least) $\frac{4}{27}$ fraction of the instances, it must find the best arm with probability at least $\frac{5}{6}$. This is again a simple Markov bound argument, as otherwise the success probability is less than

$$\frac{4}{27} \cdot \frac{5}{6} + \frac{23}{27} = \frac{79}{81}.$$

Let $n' = \frac{n}{3}$, we can show that the memory of the algorithm is $\frac{n}{24} - 1 \leq \frac{n'-2}{8}$. Furthermore, we have $\Delta_{[2]} = \beta$ for any of the instances. Therefore, the distribution in Theorem 2 reduces to this $\frac{4}{27}$ fraction of instances with $n' = \frac{n}{3}$. This implies the worst-case number of arm pulls for the distribution is at least $\Omega\left(\frac{n'}{\beta^2}\right) = \Omega\left(\frac{n}{\beta^2}\right)$. Finally, by using the fact that $\sum_{i=2}^{n} \frac{1}{\Delta_{[i]}^2} \log\log\left(\frac{1}{\Delta_{[i]}}\right) \leq \frac{1}{\beta^2} + \frac{n-1}{(\frac{1}{6})^2}$, we conclude the number of arm pulls is $\omega\left(\sum_{i=2}^{n} \frac{1}{\Delta_{[i]}^2} \log\log\left(\frac{1}{\Delta_{[i]}}\right)\right)$.

### B.4 Proof of Theorem 4

**The high-level idea**

We now analyze the hard instance with the knowledge of all gap parameters $\{\Delta_{[i]}\}_{i=2}^{n}$. Note that for the case with the gap parameters, one can easily compute INST-complexity, and the idea to construct distributions with 'hard final arm' (as we used in Theorem 1, Theorem 2, and Theorem 3) no longer works: since the algorithm knows the INST-complexity, it can easily determine if the reward of the final arm is high by checking if the INST-complexity is sufficient on first $(n-2)$ coins. In fact, if for the optimal algorithm, the $O(\mathbf{H}_2)$ arm pulls span over $\Theta(n)$ arms, then any distribution that tries to trick the algorithm seems futile since the gap parameters $\Delta_{[i]}$ for $i > 3$ must be considerably different between the cases.

To break the above barrier, we instead construct instances in which the $O(\mathbf{H}_2)$ arm pulls is 'reserved' for a small fraction of the arms, and hide the small fraction of arms among many *similar chunks*. On the high level, our hard instance satisfies the following properties: $(i)$. With the optimal algorithm, the $O(\mathbf{H}_2)$ sample complexity are mostly paid on a small chunk of arms; $(ii)$. there are $\omega(1)$ number of chunks that could possibly contain the best arm, and all of them are very similar to the chunk that actually contains the best arm; and $(iii)$. all chunks have same gaps between the arms, which makes the gap parameters $\{\Delta_{[i]}\}_{i=2}^{n}$ looks exactly the same among all the instances in the distribution. As such, the algorithm could be tricked by not being able to know *which* chunk should it search from, and it has to pay the INST-complexity to keep the best arm among sufficiently many chunks to maintain a high enough success probability. This forces the sample complexity to become $\omega(\mathbf{H}_2)$.

**The formal proof**

We construct a distribution as follows:

---

**A hard distribution for streaming MAB with all known gap parameters $\{\Delta_{[i]}\}_{i=2}^{n}$**

1. Pick parameter $t = o(n)$ and $\beta$ such that $\frac{1}{\beta} = \frac{1}{(t/n)^2}$.

---

2. Define $\frac{n}{2t}$ sets of $t$ arms $\{S_i^{\text{high}}\}_{i=1}^{\frac{n}{2t}}$ by a modified version of DIST: For each set, $(t-1)$ arms are with reward $\frac{1}{2} + \frac{i}{3} \cdot \frac{t}{n} - \beta$; one arm is with reward $\frac{1}{2} + \frac{i}{3} \cdot \frac{t}{n}$.

3. Also define $\frac{n}{2t}$ sets of $t$ arms $\{S_i^{\text{low}}\}_{i=1}^{\frac{n}{2t}}$ by simply setting the first arm with reward $\frac{1}{2} - \frac{i-1}{3} \cdot \frac{t}{n}$ and the rest of the $(t-1)$ arms with reward $\frac{1}{2} - \frac{i-1}{3} \cdot \frac{t}{n} - \beta$.

4. Sample $i^*$ uniformly at random from $[\frac{n}{2t}]$. Pick the first $i^*$ sets from $\{S_i^{\text{high}}\}_{i=1}^{\frac{n}{2t}}$ as the first $(i^* \cdot t)$ arms of the stream.

5. Pick the first $i' = \frac{n}{2t} - i^*$ sets of arms from $\{S_i^{\text{low}}\}_{i=1}^{\frac{n}{2t}}$ and appends them to the stream followed by the $(i^* \cdot t)$ arms in step 4.

6. For the rest of the arms (which is always $\frac{n}{2}$), set the their rewards as $\frac{1}{2} - \frac{i'}{3} \cdot \frac{t}{n}$.

We first make some key observations on the distribution.

(i). The quantity of gap parameters $\{\Delta_{[i]}\}_{i=2}^n$ are always the same in every instance of the distribution. To see this, note that the gaps $\Delta_{[2]}$ to $\Delta_{[t]}$ are always $\beta$. Observe that the best arm is with reward $\frac{1}{2} + \frac{i^*}{3} \cdot \frac{t}{n}$, and the rewards from the second-best to the $t$-th best arms are $\frac{1}{2} + \frac{i^*}{3} \cdot \frac{t}{n} - \beta$. The $(t+1)$-th best arm is the arm with the highest reward from the previous set, which is $\frac{1}{2} + \frac{i^*-1}{3} \cdot \frac{t}{n}$ (this is much smaller than $\frac{1}{2} + \frac{i^*}{3} \cdot \frac{t}{n} - \beta$ by the choice of $\beta$). Among this set, the gap $\Delta_{[t+1]}$ is $\frac{t}{3n}$, and the gaps $\Delta_{[t+2]}$ to $\Delta_{[2t]}$ are all $\frac{t}{3n} + \beta$. With the arms we place to the instances from $S^{\text{high}}$ and $S^{\text{low}}$, the gaps are kept the same among all instances in the distribution in the same manner till $\Delta_{[n/2]}$. The last $\frac{n}{2}$ arms are with gap $\frac{1}{3} \cdot (i^* + i') \cdot \frac{t}{n} = \frac{1}{6}$ by line 6.

(ii). For every $i^* \in [\frac{n}{2t}]$ instance in the distribution, the quantity of $\mathbf{H}_2$ is identical. This is a directly corollary of $(i)$.

(iii). The number of INST-complexity from the arm pulls on the set $S_{i^*}^{\text{high}}$ (that contains the best arm) takes at least $\frac{1}{2}$ fraction overall INST-complexity. To see this, note that the number of INST-complexity on $S_{i^*}^{\text{high}}$ is $\Omega\left(\frac{t}{\beta^2}\right) = \Omega(\frac{n^4}{t^3})$. For all the other arms, the gap is at least $\frac{t}{3n}$. Therefore, the number of arm pulls reserved for this part is at most $O\left(n \cdot \frac{1}{(t/3n)^2}\right) = O(\frac{n^3}{t^2})$. Hence, by the choice of $t = o(n)$, for sufficiently large $n$, the number of INST-complexity on $S_{i^*}^{\text{high}}$ takes at least $\frac{1}{2}$ of all the INST-complexity.

We now study the complexity of any deterministic algorithms with a memory of $\frac{t}{8}$ arms over the above distribution. We again use the Markov bound to show that for the algorithm to find the best arm with probability at least $\frac{8}{9}$, on $\frac{2}{3}$ of the instances, the algorithm has to finds the best arm with probability at least $\frac{5}{6}$, since the success probability for otherwise is less than

$$\frac{2}{3} \cdot \frac{5}{6} + \frac{1}{3} = \frac{8}{9}.$$

Note that $i^*$ is sampled uniformly at random. Therefore, we can pick $\frac{2}{3}$ of the instances by picking all the instances with $1 \leq i^* \leq \frac{2}{3} \cdot \frac{n}{2t}$. We now apply another Markov bound to show that for the algorithm to find the best arm with probability at least $\frac{5}{6}$, a $\frac{1}{2}$ fraction of them has to be solved with a probability of at least $\frac{2}{3}$. Therefore, we can calculate that for at least $\frac{2}{3} \cdot \frac{1}{2} = \frac{1}{3}$ of the choices of $i^*$, the algorithm has to at least store the best arm among $S_{i^*}^{\text{high}}$ with probability at least $\frac{2}{3}$.

Since the algorithm only has a memory of $\frac{t}{8}$ arms, by Corollary 3.3, to store the best arm with probability at least $\frac{2}{3}$, it has to use at least $\Omega\left(\frac{t}{\beta^2}\right) = \Omega(\mathbf{H}_2)$ arm pulls (per the observation $(iii)$). The algorithm is deterministic, and by observation $(i)$, the gap parameters $\{\Delta_{[i]}\}_{i=2}^n$ does not reveal any information about $i^*$. As such, the algorithm has to store the best arm for all the instances with $\frac{1}{3}$ choice of $i^*$. Hence, it has to use at least $\Omega\left(\frac{n}{t} \cdot \mathbf{H}_2\right)$ arm pulls. By the choice that $t = o(n)$, the number of arm pulls is $\omega(\mathbf{H}_2)$.

## C    Proofs of Section 3 – The Arm Trapping Lemma

### C.1    Proof of Lemma 3.1

We first assume Lemma 3.2 holds, and prove Lemma 3.1 by a reduction argument. Concretely, we show that if there exists such an algorithm (we call it `ALG`) to trap the best arm in `DIST` with probability at least $\frac{2}{3}$ and $\frac{1}{1200} \cdot \frac{n}{\beta^2}$ arm pulls, it implies an algorithm that determines the reward of the arm in Lemma 3.2 with probability at least $\frac{7}{12}$ and less than $\frac{1}{144} \cdot \frac{1}{\beta^2}$ arm pulls, which is impossible by the statement of Lemma 3.2.

The reduction is given as follows: suppose such an `ALG` to trap the best arm exists, then we design an algorithm `ALG'` to determine the reward of the special arm (denote it as $\widetilde{\text{arm}}$) as follows.

- `ALG'` samples $i^\star \in [n]$ uniformly at random, and puts $\widetilde{\text{arm}}$ at position $i^\star$.
- `ALG'` creates $(n-1)$ 'dummy arms' with rewards $(\frac{1}{2} - \beta)$, and puts them into the positions $\text{arm}_1, \cdots, \text{arm}_{i^\star-1}, \text{arm}_{i^\star+1}, \cdots, \text{arm}_n$; note that any arm pulls on these arms does *not* count towards the sample complexity on $\widetilde{\text{arm}}$.
- `ALG'` runs `ALG` with the instance (*without revealing* $i^\star$), and output with the following rules
   1. If `ALG` makes more than $\frac{1}{1200} \cdot \frac{n}{\beta^2}$ arm pulls at any time, terminate and output $\widetilde{p} = \frac{1}{2} - \beta$;
   2. Else, if `ALG` makes more than $\frac{1}{150} \cdot \frac{1}{\beta^2}$ arm pulls on $\widetilde{\text{arm}}$, terminate and output $\widetilde{p} = \frac{1}{2}$;
   3. Otherwise (if `ALG` makes at most $\frac{1}{150} \cdot \frac{1}{\beta^2}$ arm pulls on $\widetilde{\text{arm}}$), output with the following rules
      - $\widetilde{p} = \frac{1}{2}$ if $\widetilde{\text{arm}}$ is among the arms returned by `ALG`.
      - $\widetilde{p} = \frac{1}{2} - \beta$ if $\widetilde{\text{arm}}$ is *not* among the arms returned by `ALG`.

It is straightforward to see that the number of arm pulls on $\widetilde{\text{arm}}$ is at most $\frac{1}{150} \cdot \frac{1}{\beta^2}$, as we explicitly terminate the algorithm once the sample complexity exceeds this limit.

We now verify that `ALG'` determines the reward of the arm with probability at least $\frac{7}{12}$. We first show that if $p = \frac{1}{2}$, the error probability is at most $\frac{1}{3}$. To see this, note that for `ALG'` to make mistakes when $p = \frac{1}{2}$, `ALG` needs to either makes more than $\frac{1}{1200} \cdot \frac{n}{\beta^2}$ arm pulls, or fail to include $\widetilde{\text{arm}}$ among the arm it stores. By the assumption of the algorithm, the probability for either of the cases to happen is at most $\frac{1}{3}$.

On the other hand, when $p = \frac{1}{2} - \beta$, we show that the error probability is at most $\frac{1}{2}$. To see this, note that when $p = \frac{1}{2} - \beta$, `ALG'` could make mistakes on the following occasions.

- `ALG` fails (returns arbitrary results, but terminates with $\frac{1}{1200} \cdot \frac{n}{\beta^2}$ arm pulls).
- `ALG` returns correctly, but it also includes $\widetilde{\text{arm}}$.
- `ALG` returns correctly, but uses more than $\frac{1}{40} \cdot \frac{1}{\beta^2}$ arm pulls on $\widetilde{\text{arm}}$ such that `ALG'` falsely terminates and returns $p = \frac{1}{2}$.

The probability for the first scenario to happen is clearly at most $\frac{1}{3}$. For the second scenario, note that $\widetilde{\text{arm}}$ has the same reward as other arms, and they become identical random variables. Therefore, we have

$$\Pr\left( \widetilde{\text{arm}} \text{ included by ALG} \mid p = \frac{1}{2} - \beta \right) = \Pr\left( \text{arm}_j \text{ included by ALG} \mid p_j = \frac{1}{2} - \beta \right)$$
$$= \Pr\left( \text{arm}_j \text{ included by ALG}, \forall j \in [n] \right)$$

Since `ALG` only returns $\frac{1}{8}$ fraction of the arms, the probability for $\widetilde{\text{arm}}$ to be falsely included to the output is $\frac{1}{8}$.

For the third scenario, we show that conditioning on $p = \frac{1}{2} - \beta$, it is with lower probability that `ALG` excessively makes arm pulls on $\widetilde{\text{arm}}$ while keeping the overall number of arm pulls low. By an argument that is similar to the previous scenario, the number of arm pulls on $\widetilde{\text{arm}}$ is an identical random variable with the number of arm pulls on any of the other arms. Therefore, denote $T_{i^\star}$ as the

number of arm pulls on $\widetilde{\mathsf{arm}}$ and $T_j$ the number of arm pulls on any arm $j$, we have

$$\mathbb{E}\left[T_{i^\star}\,\Big|\, p = \frac{1}{2} - \beta\right] = \mathbb{E}\left[T_j\,\Big|\, p_j = \frac{1}{2} - \beta, \forall j \in [n]\right]$$

$$= \mathbb{E}\left[T_j\right], \ \forall j \in [n]$$

Hence, the number of arm pulls on $\widetilde{\mathsf{arm}}$ under $p = \frac{1}{2} - \beta$ equals to the expectation on each arm (the *average*). More formally, we have

$$\mathbb{E}\left[\text{number of total arm pulls of ALG}\,\Big|\, p = \frac{1}{2} - \beta\right] = \mathbb{E}\left[\sum_{j=1}^{n} T_j\,\Big|\, p_j = \frac{1}{2} - \beta\right]$$

$$= \sum_{j=1}^{n} \mathbb{E}\left[T_j\,\Big|\, p_j = \frac{1}{2} - \beta\right]$$

(linearity of expectation)

$$= n \cdot \mathbb{E}\left[T_{i^\star}\,\Big|\, p_{i^\star} = \frac{1}{2} - \beta\right].$$

(identical random variables with equal expectations)

That is to say, conditioning on $p = \frac{1}{2} - \beta$, we have $\mathbb{E}\left[T_{i^\star}\,|\,p = \frac{1}{2} - \beta\right] = \frac{\mathbb{E}[\text{number of total arm pulls of ALG}]}{n} = \frac{1}{1200} \cdot \frac{1}{\Delta^2}$ arm pulls. By Markov bound, the probability for the number of arm pulls on $\widetilde{\mathsf{arm}}$ to be more than $\frac{1}{150} \cdot \frac{1}{\Delta^2}$ is at most $\frac{1}{8}$. Therefore, by summarizing the above scenarios, we conclude that the probability for ALG' to make mistake under $p = \frac{1}{2} - \beta$ is bounded by $\frac{1}{3} + \frac{2}{3} \cdot (\frac{1}{8} + \frac{1}{8}) = \frac{1}{2}$.

Summarizing the above two cases on $p$ gives us the failure probability of at most

$$\Pr\left(\text{ALG' fails}\right) = \frac{1}{2} \cdot \Pr\left(\text{ALG' fails} \mid p = \frac{1}{2}\right) + \frac{1}{2} \cdot \Pr\left(\text{ALG' fails} \mid p = \frac{1}{2} - \beta\right)$$

$$= \frac{1}{2} \cdot \frac{1}{3} + \frac{1}{2} \cdot \frac{1}{2} = \frac{5}{12}.$$

As such, ALG' can determine the reward of the arm in Lemma 3.2 with probability at least $\frac{7}{12}$ and $\frac{1}{150} \cdot \frac{1}{\Delta^2}$ arm pulls, which contradicts the lemma itself. Therefore, the lower bound is obtained.

**Remark C.1.** Note that it is possible to prove Lemma 3.1 without the reduction argument. For instance, one can use the approaches developed by [28, 14] to obtain a similar result. The merit of our proof is that it provides a *black-box* way to prove MAB lower bounds without resorting to previous techniques. Furthermore, the idea to prove lower bounds by 'embedding' a single instance into multiple copies and using the 'direct sum' argument is extensively used in the theoretical computer science literature. Our proof shows that such an idea works for machine learning lower bounds, which gives an interesting connection between the areas.

### C.2 Proof of Lemma 3.2

We now turn to the skipped proof of Lemma 3.2. The lemma is a standard result that has been shown by multiple previous work [33, 1], but none of them could provide a blackbox lemma for our case to use. Therefore, we provide an information-theoretic proof for Lemma 3.2. We reach the conclusion by showing that to distinguish two distributions with a bounded KL divergence with a sufficiently high probability, a certain number of samples are necessary. En route to the proof, we begin with showing the error lower bound as a function of the total variation distance, thus establishing the first version of connection between the error and the divergence. We then use Pinsker's inequality (proposition C.3) to transform the total variation distance to the functions of KL divergence. Furthermore, by leveraging the chain rule and the independence between samples, we transform the KL divergence of the distribution with $m$ samples to $m$ copies of the KL divergence of the distribution with a single sample. The latter quantity can be bounded as $O(\beta^2)$ since the reward is from Bernoulli distributions. And finally, the sample lower bound can be obtained by solving the inequality that upper-bounds the error probability.

Before showing the proof, we list the technical tools we are going to use. The first technical tool is the standard KL divergence for discrete distributions:

**Proposition C.2.** *Let two discrete distributions $X$ and $Y$ be on the same set of supports $\mathcal{S}$, the KL divergence between $X$ and $Y$ is defined as*

$$D_{KL}(X||Y) = \sum_{s \in \mathcal{S}} X(s) \log \left( \frac{X(s)}{Y(s)} \right).$$

To connect the total variation distance with the KL divergence, we use Pinker's inequality:

**Proposition C.3** (Pinker's inequality). *Let $\delta_{TV}(\cdot, \cdot)$ denote the total variation distance and $D_{KL}(\cdot||\cdot)$ denote the KL divergence. For two distributions $X$ and $Y$, we have*

$$\delta_{TV}(X, Y) \leq \sqrt{\frac{1}{2} \cdot D_{KL}(X||Y)}.$$

We now formalize the above intuitions. To begin with, we present the following claim

**Claim C.4.** *Let $X$ and $Y$ be two distributions, and $P$ is determined by*

$$P = \begin{cases} X, & \text{w.p. } \frac{1}{2}; \\ Y, & \text{w.p. } \frac{1}{2}. \end{cases}$$

*Any algorithm that samples once from $P$ to determine which distribution does $P$ follows makes an error of at least $\frac{1}{2} \cdot (1 - \delta_{TV}(X, Y))$.*

*Proof.* We slightly abuse the notations of $X$ and $Y$ to let them denote both the distributions and the mass functions. Let $\hat{P}$ be the prediction of the algorithm, we can write the error as

$$\Pr(\text{error}) = \Pr(\hat{P} = X|P = Y) \cdot \Pr(P = Y) + \Pr(\hat{P} = Y|P = X) \cdot \Pr(P = X)$$
$$= \frac{1}{2} \cdot (\Pr(\hat{P} = X|P = Y) + \Pr(\hat{P} = Y|P = X)).$$

For discrete distributions, the conditional error terms are no worse than the total probability of the 'converse likelihoods.' That is, the total probability of the samples that by evaluating on the supports are from one distribution, but actually from another. More formally, we have $\Pr(\hat{P} = X|P = Y) \geq \Pr(X(s) > Y(s)|s \sim Y)$, and $\Pr(\hat{P} = Y|P = X) \geq \Pr(Y(s) > X(s)|s \sim X)$. Therefore, the error is lower bounded by

$$\Pr(\text{error}) \geq \frac{1}{2} \cdot (\Pr(X(s) > Y(s)|s \sim Y) + \Pr(X(s) < Y(s)|s \sim X))$$
$$= \frac{1}{2} \cdot ( \sum_{s:X(s) \geq Y(s)} Y(s) + \sum_{s:X(s) < Y(s)} X(s))$$
$$= \frac{1}{2} \cdot ( \sum_{s:X(s) \geq Y(s)} Y(s) + 1 - \sum_{s:X(s) \geq Y(s)} X(s))$$
$$= \frac{1}{2} \cdot (1 - \sum_{s:X(s) \geq Y(s)} (X(s) - Y(s)))$$
$$= \frac{1}{2} \cdot (1 - \delta_{TV}(X, Y)),$$

which is as desired. $\square$

Note that Claim C.4 is applicable to every pairs of distributions, even the distribution with multiple samples. Therefore, if we use $X^{[m]}$ and $Y^{[m]}$ to denote the distribution of $m$ samples from $X$ and $Y$, we can write the error probability lower bound as

$$\Pr(\hat{P} \neq P) \geq \frac{1}{2} \cdot \left(1 - \delta_{TV}(X^{[m]}, Y^{[m]})\right).$$

To provide a stronger bound, we transform the above inequality to a function of the KL-divergence. Note that with Pinker's inequality (Proposition C.3), there is $\delta_{TV}(X^{[m]}, Y^{[m]}) \leq$

$\sqrt{\frac{1}{2} \cdot D_{\mathrm{KL}}(X^{[m]}||Y^{[m]})}$. Since $X^{[m]}$ and $Y^{[m]}$ are from independent identically samples, for any of the marginal distribution function (denote as $X^i$), we have $X^1 = X^2 = \cdots = X^m = X$. Therefore, we have

$$
\begin{aligned}
D_{\mathrm{KL}}(X^{[m]}, Y^{[m]}) &= D_{\mathrm{KL}}\Big(X^m||Y^m\Big) + D_{\mathrm{KL}}\Big((X^{[m-1]}|X^m)||(Y^{m-1}|Y^m)\Big) \quad \text{(by the chain rule)} \\
&= D_{\mathrm{KL}}\Big(X||Y\Big) + D_{\mathrm{KL}}\Big(X^{[m-1]}||Y^{m-1}\Big) \\
&\qquad\qquad \text{(by the independence and identity of the random variables)} \\
&= \cdots\cdots\cdots \\
&= m \cdot \Big(D_{\mathrm{KL}}(X||Y)\Big).
\end{aligned}
$$

Since both $X$ and $Y$ are Bernoulli random variables, we have

$$
\begin{aligned}
\Big(D_{\mathrm{KL}}(X||Y)\Big) &= \sum_i \Pr(X = i) \log\big(\frac{\Pr(X = i)}{\Pr(Y = i)}\big) \\
&= (\frac{1}{2} + \beta) \cdot \log(1 + 2\beta) + (\frac{1}{2} - \beta) \cdot \log(1 - 2\beta) \\
&= \frac{1}{2} \cdot \log((1 + 2\beta)(1 - 2\beta)) + \beta \cdot \log(\frac{1 + 2\beta}{1 - 2\beta}) \\
&\leq \beta \cdot \log(\frac{1 + 2\beta}{1 - 2\beta}) \qquad\qquad\qquad\quad (\log(1 - 4\beta^2) < 0) \\
&\leq \beta \cdot \log(2^{6 \cdot \beta}) \qquad\qquad\qquad (\tfrac{1+2\beta}{1-2\beta} \leq 2^{6 \cdot \beta} \text{ for } \beta \in (0, \tfrac{1}{6})) \\
&= 6 \cdot \beta^2
\end{aligned}
$$

Therefore, we have

$$
\begin{aligned}
\Pr(\hat{P} \neq P) &\geq \frac{1}{2} \cdot \Big(1 - \sqrt{\frac{1}{2} \cdot D_{\mathrm{KL}}(X^m, y^m)}\Big) \\
&\geq \frac{1}{2} \cdot \Big(1 - \sqrt{\frac{1}{2} \cdot 6m \cdot \beta^2}\Big) \\
&\geq \frac{1}{2} \cdot \Big(1 - 2 \cdot \beta \cdot \sqrt{m}\Big).
\end{aligned}
$$

On the other hand, we want the error probability to be at most $\frac{5}{12}$, which means $\frac{1}{2} \cdot \Big(1 - 2\beta \cdot \sqrt{m}\Big) \leq \frac{5}{12}$, which solves to $m \geq \frac{1}{144} \cdot \frac{1}{\beta^2}$.

**Remark C.5.** We pick $\beta < \frac{1}{6}$ to obtain a clean constant for the number of samples, and it is sufficient to prove the lower bounds to follow. In fact, if one intends to generalize the range of $\beta$, one can always decease the multiplicative term up to a constant factor, and relax the inequality used in the derivative of the upper bound of the KL divergence. Nonetheless, we do not aim to get the best possible range in this paper.

**Remark C.6.** Very recently, [2] obtains an arm-trapping lower bound that is similar-in-spirit to ours. However, we remark that their bound focused on the much stronger scenario that the algorithm does not learn the *distribution* of the arms (as opposed to simply trap the arm); and their sample complexity lower bound is much weaker – it has no dependency on $n$.

# D  Missing Analysis of the Algorithm

## D.1  A High-level Overview of the Analysis

Before showing the formal analysis, we outline the high-level intuitions of the analysis below (focusing on the case $\delta = \Theta(1)$). Any multi-level challenging rules have to fulfill three properties: *soundness*, as in the best arm should be able to become the king; *completeness*, as in after the best arm becomes the king, it should not be discarded; and *sample complexity*, which is desired to be proportional to the INST-complexity $\mathbf{H}_2^\delta$.

Let us first consider the scenario that arm$^*$ is already the king, under which our concerns are focused on completeness and sample complexity. One way to guarantee the completeness is to pull both the king and the challenger $O(1/\Delta_{[2]}^2)$ times at the first level, and increase the number of arm pulls geometrically in the levels afterwards. The correctness of such a method was proved by [5] using a random walk argument. Nonetheless, with such a mechanism, the sample complexity could be as high as $\Theta(n/\Delta_{[2]}^2)$, which is clearly not instance-sensitive.

To tackle this issue, we design a new *instance-sensitive challenge subroutine*, inspired by [26]. The subroutine maintains a 'guess' $\tilde{\Delta}_\ell$ of the gap between the king and the challenger arm. The parameter $\tilde{\Delta}_\ell$ starts with a sufficiently large value to avoid excessive arm pulls, and updates itself by decreasing the value with a constant factor as the level $\ell$ goes up. After $\tilde{\Delta}_\ell$ becomes smaller than the real $\Delta_{[i]}$, the probability for the challenger arm to defeat arm$^*$ becomes very small, and we can show that the sample complexity of the challenge (from this single arm) is $O\left(\frac{1}{\Delta_{[i]}^2}\log\log(\frac{1}{\Delta_{[i]}})\right)$ *in expectation*.

We now generalize to the scenario that *any* arm could be the king. The strategy we used in the previous case no longer works here, as the soundness immediately becomes an issue: the rule requires the challenger to defeat the king at every level, but arm$^*$ might not be able to defeat an arm with a close reward by a large guess of $\tilde{\Delta}_\ell$. Furthermore, if some other arm besides arm$^*$ becomes the king, there is no guarantee on the sample complexity.

A remedy for the former issue is the new notion of *epochs*: we refer to the period in the stream that a fixed arm remains the king as an epoch and throughout the epoch, maintain a *fixed* estimate $p_{\sf est}$ as an approximation of the reward of the *best arm* $p^*$ (not necessarily the empirical reward of the current king). We then require each challenger to win only against $(p_{\sf est} - \frac{\tilde{\Delta}}{4})$ – an easier rule that arm$^*$ is able to satisfy at *all* levels upon arrival. Moreover, the $p_{\sf est}$ value itself starts as $\frac{\Delta_{[2]}}{20}$ and is increased by at least $\frac{\Delta_{[2]}}{20}$ whenever the king is defeated (which implies the guess for $p_{\sf est}$ was not sufficiently large). This allows us to control the number of all kings that are ever defeated to $O(1/\Delta_{[2]})$ – the importance of this will be clear once we address the second challenge.

The second challenge requires a more careful treatment. In fact, this is where we need the assumptions of the random arrival and the knowledge of the INST-complexity $\mathbf{H}_2^\delta$. Our general approach is to give a *budget* to each king for performing the arm pulls in the challenges, and discard the king whenever it exhausts its budget – the idea of budgeting also appears in [5] but our analysis of budget is entirely different than theirs. From an amortized perspective, we can give the king a budget of roughly $O(\mathbf{H}_2^\delta/n)$ for each arriving arm to bound the sample complexity by $O(\mathbf{H}_2^\delta)$. Unfortunately, this on its own does not guarantee the completeness: while arm$^*$ as a king recieves a budget that is sufficient *in expectation*, since the $\Delta_i$-value of different challengers can be *highly variant*, if the arms come in an adversarial manner, arm$^*$ could quickly exhaust its budget. Therefore, we further need the condition of the random arrival of the arms. Still, even this way, the variance in budget use does not immediately drop sufficiently. However, we can now show that after $O(\text{poly}(\log n/\Delta_{[2]}))$ arms, the number of arm pulls will fall below the budget with a high (constant) probability. For the algorithm to 'warm-up' with these starting arms, we can directly call the GAME-OF-COINS algorithm of [5] to guarantee the correctness, which will pay an $\text{poly}(\log n/\Delta_{[2]})$ overhead on sample complexity. Finally, since we also managed to limit the total number of kings to be ever defeated by $O(1/\Delta_{[2]})$, the overhead applied to *all* the kings is also bounded by $\text{poly}(\log n/\Delta_{[2]})$.

### D.2 The Formal Analysis

We provide the formal analysis of the algorithm in this section. We start with bounding the sample complexity of the algorithm.

**Lemma D.1** (**Sample Complexity**). *The total number of arm pulls used by the algorithm is*

$$O\left(\mathbf{H}_2^\delta + \frac{1}{\Delta_{[2]}^7}\cdot\log(\frac{n}{\delta})^2\cdot\log^2\left(\frac{1}{\delta}\log\left(\frac{1}{\Delta_{[2]}}\right)\right)\right).$$

*Proof.* There are three sources of arm pulls: the GAME-OF-COINS algorithm, the challenge subroutine, and the arm pulls to update $p_{\sf est}$. For the first part, one can directly use Proposition A.4 to bound

the sample complexity as

$$O\left(\frac{n'}{(\Delta_{[2]}/4)^2} \cdot \log(\frac{1}{\delta})\right) = O\left(\frac{1}{\Delta_{[2]}^7} \cdot \log(\frac{n}{\delta})^2 \cdot \log^2\left(\frac{1}{\delta}\log\left(\frac{1}{\Delta_{[2]}}\right)\right)\right).$$

Note that the best arm may *not* be among the early arms; but since the GAME-OF-COINS algorithm only adds a budget of $O(\frac{1}{\Delta_{[2]}^2})$ for each arriving arm, the sample complexity is bounded.

For the challenge subroutine, note that we only collect a budget of $b = O(\frac{\mathbf{H}_2^\delta}{n})$ for each arriving arm so the total budget is at most $O(\mathbf{H}_2^\delta)$. Each unit of budget is responsible for two arm pulls. As a result, the the sample complexity of the challenge subroutine throughout the algorithm is $O(\mathbf{H}_2^\delta)$. And finally, if king is discarded, the algorithm pays another $O(\frac{1}{\Delta_{[2]}^2} \cdot \log(\frac{n}{\delta}))$ arm pulls. The total number of times a king can be discarded is at most $\frac{20}{\Delta_{[2]}}$ (as after that $p_{\mathsf{est}} > 1$), which means this part pays $O\left(\frac{\log(n/\delta)}{\Delta_{[2]}^3}\right)$ arm pulls, that is suppressed in the asymptotic sample complexity. $\qquad\square$

**Lemma D.2 (Correctness).** *With probability at least $(1 - \delta)$, the algorithm outputs the the best arm* arm*.

To prove the lemma, we need to show both the soundness and the completeness. To do so, we need the following key claims on the behavior of our instance-sensitive challenge subroutine. Considering the proofs are technical, we postpone them to the appendix—however, we shall note that the entire idea behind the algorithm hinges on these claims.

**Claim D.3.** *Assuming $p_{\mathsf{est}} \geq p^* - \frac{\Delta_{[2]}}{2}$, the expected number of arm pulls of the* instance-sensitive *challenge subroutine on* arm$_{[i]}$ *is at most* $\left(600 \cdot \frac{1}{\Delta_{[i]}^2} \log\left(\frac{1}{\delta}\log\left(\frac{1}{\Delta_{[i]}^2}\right)\right)\right)$.

*Proof.* Define $\ell_{\Delta_i} := \left\lceil \log(\frac{1}{\Delta_{[i]}}) \right\rceil$. We analyze the number of arm pulls of arm$_{[i]}$ for the cases of $\ell \leq \ell_{\Delta_i}$ and $\ell > \ell_{\Delta_i}$, respectively. For all the levels $\ell \leq \left\lceil \log(\frac{1}{\Delta_{[i]}}) \right\rceil$, the overall number of arm pulls is at most:

$$\sum_{\ell=1}^{\ell_{\Delta_i}} s_\ell = \sum_{\ell=1}^{\ell_{\Delta_i}} \frac{4}{\tilde{\Delta}^2} \cdot \log(\frac{1}{\delta_\ell}) = \sum_{\ell=1}^{\ell_{\Delta_i}} 4 \cdot 16 \cdot (2^{2(\ell-1)}) \cdot \log(\frac{50 \cdot \ell^3}{\delta})$$

$$\leq 4 \cdot 16 \cdot 2^{2 \cdot \log(\frac{1}{\Delta_{[i]}})} \cdot \log(\frac{50 \cdot \log(\frac{1}{\Delta_{[i]}})^3}{\delta}) \leq 576 \cdot \frac{1}{\Delta_{[i]}^2} \cdot \log\left(\frac{1}{\delta}\log\left(\frac{1}{\Delta_{[i]}}\right)\right).$$

$$(\log(50^{\frac{1}{3}}) < 2)$$

For the levels $\ell > \ell_{\Delta_i}$, we show that the probability for arm$_{[i]}$ to survive level $\ell$ is at most $\prod_{\ell'=\ell_{\Delta_i}+1}^{\ell} \delta_{\ell'}$. Note that for $\ell > \ell_{\Delta_i}$, there is $\tilde{\Delta} \leq \frac{1}{4} \cdot \Delta_{[i]}$. Since we assume $p_{\mathsf{est}} > p^* - \frac{\Delta_{[2]}}{2} > p^* - \frac{\Delta_{[i]}}{2}$, for the challenger arm to defeat the king, it should have an empirical reward of at least $p^* - \frac{3}{4} \cdot \Delta_{[i]}$. On the other hand, we know its true reward is $p^* - \frac{1}{4} \cdot \Delta_{[i]}$. By Proposition A.2, we have

$$\Pr(\widehat{p}_{[i]} \geq p^* - \frac{3}{4} \cdot \Delta_{[i]}) = \Pr(\widehat{p}_{[i]} - p \geq \frac{1}{4} \cdot \Delta_{[i]}) \leq \exp(-\frac{1}{4} \cdot 4 \cdot \log(\frac{1}{\delta_\ell})) = \delta_\ell.$$

Based on the challenging rules, the above inequality implies that

$$\Pr\left(\mathsf{arm}_{[i]} \text{ reaches level } \ell \mid \mathsf{arm}_{[i]} \text{ survives previous levels}\right) \leq \delta_\ell.$$

Therefore, defining $X_i$ as the number arm pulls for arm$_{[i]}$ under the challenge subroutine, we can bound the expectation of the total arm pulls for $\ell > \ell_{\Delta_i}$ as

$$\mathbb{E}\left[X_i \mid \ell > \ell_{\Delta_i}\right] = \sum_{\ell=\ell_{\Delta_i}+1}^{\infty} \Pr\left(\mathsf{arm}_{[i]} \text{ reaches level } \ell\right) \cdot (\text{number of total arm pulls})$$

$$\leq \sum_{\ell=\ell_{\Delta_i}+1}^{\infty} \left( \prod_{\ell'=\ell_{\Delta_i}+1}^{\ell} \delta_{\ell'} \right) \cdot 5 \cdot s_\ell \qquad\qquad (s_\ell \geq \tfrac{1}{4} \cdot \textstyle\sum_{\ell'=1}^{\ell-1} s_{\ell'})$$

$$\leq 4 \cdot 16 \cdot 9 \sum_{\ell=\ell_{\Delta_i}+1}^{\infty} \frac{1}{\Delta_{[i]}^2} \cdot 2^{2 \cdot (\ell - \ell_{\Delta_i})} \cdot \log(\tfrac{\ell}{\delta}) \cdot \left( \prod_{\ell'=\ell_{\Delta_i}+1}^{\ell} \frac{\delta}{50} \cdot \frac{1}{8} \right)$$
$$(\ell_{\Delta_i} + 1 \geq 2)$$

$$\leq 4 \cdot 16 \cdot 9 \cdot \sum_{\ell'=1}^{\infty} \frac{1}{\Delta_{[i]}^2} 2^{2\ell'} \cdot \log(\tfrac{\ell' + \ell_{\Delta_i}}{\delta}) \cdot \left( \prod_{\ell''=1}^{\ell'} \frac{\delta}{50} \cdot \frac{1}{8} \right) \quad (\text{let } \ell' = \ell - \ell_{\Delta_i})$$

$$\leq 4 \cdot 16 \cdot 9 \cdot \frac{1}{100} \cdot \frac{1}{\Delta_{[i]}^2} \cdot \sum_{\ell'=1}^{\infty} (\tfrac{1}{2})^{\ell'} \cdot (\tfrac{4}{3})^{\ell'} \cdot \log(\frac{\log(\frac{1}{\Delta_{[i]}})}{\delta}) \qquad (\tfrac{\delta}{50} \leq \tfrac{1}{100})$$

$$\leq 12 \cdot \frac{1}{\Delta_{[i]}^2} \cdot \log\left( \frac{1}{\delta} \log\left( \frac{1}{\Delta_{[i]}} \right) \right).$$

Hence, by summarizing the above results, we can get

$$\mathbb{E}\left[X_i\right] \leq \sum_{\ell=1}^{\ell_{\Delta_i}} s_\ell + \mathbb{E}\left[X_i \mid \ell > \ell_{\Delta_i}\right] \leq 600 \cdot \frac{1}{\Delta_{[i]}^2} \cdot \log\left( \frac{1}{\delta} \log\left( \frac{1}{\Delta_{[i]}} \right) \right),$$

concluding the proof. $\qquad\qquad\qquad\qquad\qquad\qquad\qquad\qquad\qquad\qquad\qquad\qquad\qquad\square$

To continue we need some definition. For any set $S$ of *consecutive* arms in the stream, define:

$$\mathsf{Ind}(S) := \left\{ j \mid \mathsf{arm}_{[j]} \in S \right\} \qquad \mathbf{H}_2^\delta(S) := \sum_{j \in \mathsf{Ind}(S)} \frac{1}{\Delta_{[j]}^2} \log\left( \frac{1}{\delta} \log\left( \frac{1}{\Delta_{[j]}} \right) \right).$$

We can use claim D.3 to show that $\mathbf{H}_2^\delta(S)$ characterizes the sample complexity of the algorithm on the arms in $S$ for any sufficiently large set $S$, as Claim D.4.

**Claim D.4.** *Assuming $p_{est} \geq p^* - \frac{\Delta_{[2]}}{2}$, for a set of consecutive arms $S \subseteq \{\mathsf{arm}_i\}_{i=1}^n$ that does not contain $\mathsf{arm}^*$ and has a size $|S| \geq m_{\mathsf{early}}$, the number of arm pulls on the set $S$ under the challenge rule is at most $\left( 900 \cdot \mathbf{H}_2^\delta(S) \right)$ w.p. at least $\left( 1 - \frac{\delta}{16n^2} \right)$.*

*Proof.* Define random variables $\{Y_j\}_{j \in \mathsf{Ind}(S)}$ as the number of arm pulls used on $\mathsf{arm}_{[j]} \in S$. Let the number of arm pulls on $S$ be $Y = \sum_{j \in \mathtt{Ind}(S)} Y_j$. By Claim D.3, one can show that

$$\mathbb{E}\left[Y\right] = \sum_{j \in \mathtt{Ind}(S)} \mathbb{E}\left[Y_j\right] \leq 600 \cdot \sum_{j \in \mathtt{Ind}(S)} \frac{1}{\Delta_{[j]}^2} \log\left( \frac{1}{\delta} \log\left( \frac{1}{\Delta_{[j]}} \right) \right) = 600 \cdot \mathbf{H}_2^\delta(S).$$

Moreover, the challenge subroutine will pull an arm at most $\theta := \frac{128}{\Delta_{[2]}^2} \cdot \log(\frac{n}{\delta})$ times at most. Thus, each $Y_j \in [1, \theta]$. As such, by Chernoff bound,

$$\Pr\left(Y \geq 900 \cdot \mathbf{H}_2^\delta(S)\right) \leq \Pr\left(Y \geq 1.5 \cdot \mathbb{E}\left[Y\right]\right) \leq \exp(-\frac{0.5^2 \cdot \mathbb{E}\left[Y\right]}{3\theta}) \leq \exp(-\frac{0.5^2 \cdot m_{\mathsf{early}}}{3\theta})$$

$$= \exp(-\frac{1}{12} \cdot \frac{\frac{30}{\Delta_{[2]}^4} \cdot \log^2(\frac{n}{\delta}) \cdot \log^2\left( \frac{1}{\delta} \log\left( \frac{1}{\Delta_{[2]}} \right) \right)}{3 \cdot \frac{128}{\Delta_{[2]}^2} \cdot \log(\frac{n}{\delta})})$$

$$\leq \exp(-4 \cdot \log(\frac{n}{\delta})) \leq \frac{\delta}{16n^2},$$

as desired. $\qquad\qquad\qquad\qquad\qquad\qquad\qquad\qquad\qquad\qquad\qquad\qquad\qquad\qquad\qquad\square$

We next use the random arrival assumption to prove $\mathbf{H}_2^\delta(S) \approx \frac{|S|}{n} \cdot \mathbf{H}_2^\delta$ for all sufficiently large $S$.

**Claim D.5.** *Assuming the arms arrive in a random order, for any set of consecutive arms $S$ with size $|S| \geq m_{\text{early}}$, we have $\mathbf{H}_2^\delta(S) \leq 2 \cdot |S| \cdot \frac{\mathbf{H}_2^\delta}{n}$ with probability at least $(1 - \frac{\delta}{16n^2})$.*

*Proof.* Fix indices of any set $S$ of consecutive arm. For any $i \in S$, define $Z_i$ as the random variable that takes value $\frac{1}{\Delta_{[j]}^2} \log(\frac{1}{\delta} \cdot \log \frac{1}{\Delta_{[j]}})$ where $\mathsf{arm}_{[j]}$ is the arm arriving in the position $i$ of $S$. Define $Z := \sum_{i \in S} Z_i$ and note that $\mathbf{H}_2^\delta(S) = Z$. Moreover, by the randomness of the stream,

$$\mathbb{E}\left[Z\right] = \sum_{i \in S} \mathbb{E}\left[Z_i\right] = \sum_{i \in S} \frac{1}{n} \cdot \sum_{j=1}^{n} \frac{1}{\Delta_{[j]}^2} \log(\frac{1}{\delta} \cdot \log \frac{1}{\Delta_{[j]}}) = \frac{|S|}{n} \cdot \mathbf{H}_2^\delta.$$

Define $\theta := \frac{1}{\Delta_{[2]}^2} \log(\frac{1}{\delta} \cdot \log \frac{1}{\Delta_{[2]}})$ and note that each $Z_i \in [1, \theta]$ with this definition. Considering the distribution of $Z$ is sampling without replacement, by Proposition A.3, we have that

$$\Pr(Z - \mathbb{E}\left[Z\right] \geq \mathbb{E}\left[Z\right]) \leq \exp\left(-\frac{2 \cdot \mathbb{E}\left[Z\right]^2}{|S| \cdot \theta^2}\right) \leq \exp\left(-\frac{|S|}{\theta^2}\right)$$

$$\leq \exp\left(-\frac{\frac{30}{\Delta_{[2]}^4} \cdot \log^2(\frac{n}{\delta}) \cdot \log^2\left(\frac{1}{\delta} \log\left(\frac{1}{\Delta_{[2]}}\right)\right)}{(\frac{1}{\Delta_{[2]}^2} \log(\frac{1}{\delta} \cdot \log \frac{1}{\Delta_{[2]}})^2}\right)$$

$$(\text{$|S| \geq m_{\text{early}}$ and by the choice of $m_{\text{early}}$})$$

$$= \exp\left(-30 \cdot (\log(n/\delta))^2\right) \ll \frac{\delta}{16n^2}.$$

As such, we have that for any sufficiently large $S$ as specified by the claim, $\mathbf{H}_2^\delta(S) \leq 2 \cdot |S| \cdot \frac{\mathbf{H}_2^\delta}{n}$ with probability at least $(1 - \frac{\delta}{16n^2})$, concluding the proof. $\square$

As a direct result of claims D.4 and D.5, we have:

**Claim D.6.** *Assuming the arms arrive in a random order and $p_{\text{est}} \geq p^* - \frac{\Delta_{[2]}}{2}$, the number of arm pulls used by the challenge subroutine on every set $S$ of consecutive arms that does not contain $\mathsf{arm}^*$ and has size $|S| \geq m_{\text{early}}$ is at most $1800 \cdot \frac{|S|}{n} \cdot \mathbf{H}_2^\delta$ with probability at least $(1 - \frac{\delta}{8})$.*

*Proof.* There are at most $n^2$ sets of consecutive arms in total. Therefore, by a union bound on Claim D.4, the number of arm pulls used by the challenge subroutine on every set of consecutive arms is at most $900 \cdot \mathbf{H}_2^\delta(S)$ with probability at least $1 - \frac{\delta}{16}$.

By a similar argument applied to Claim D.5, we have that for every set of consecutive arms of size at least $m_{\text{early}}$, $\mathbf{H}_2^\delta(S) \leq 2 \cdot \frac{|S|}{n} \cdot \mathbf{H}_2^\delta$ with probability at least $1 - \frac{\delta}{16}$. Under the condition that both of the above happen, which is with probability at least $(1 - \frac{\delta}{8})$ by a union bound, the umber of arm pulls used by the challenge subroutine on every set of consecutive arms with size at least $m_{\text{early}}$ and without $\mathsf{arm}^*$ is at most $1800 \cdot \frac{|S|}{n} \cdot \mathbf{H}_2^\delta$ as desired. $\square$

We are now ready to show the correctness of the algorithm.

**Lemma D.7 (Soundness).** *W.p. at least $1 - \frac{\delta}{2}$, $\mathsf{arm}^*$ becomes the king upon arrival.*

*Proof.* We first show that with probability at least $1 - \frac{\delta}{4}$, the estimation $p_{\text{est}}$ is at most $p^* - \frac{\Delta_{[2]}}{4}$ before $\mathsf{arm}^*$ arrives. Since $p_{\text{est}}$ is updated as maximum of $\widehat{p}_{\text{challenger}}$ and $p_{\text{est}} + \frac{\Delta_{[2]}}{20}$ in Line (2) of the algorithm, we analyze these two arguments of the max-term separately.

- For $p_{\text{est}} \geq p^* - \frac{\Delta_{[2]}}{4}$ to happen, the empirical reward of the challenger should be $\widehat{p}_{\text{challenger}} \geq p^* - \frac{\Delta_{[2]}}{4} - \frac{\Delta_{[2]}}{20} \geq p^* - \frac{\Delta_{[2]}}{2}$. But as $\mathsf{arm}^*$ has not arrived yet, the probability for any other arm to achieve such a high empirical reward in Line (1) is at most $\frac{\delta}{8n}$ by Proposition A.2. Thus, w.p. at least $(1 - \frac{\delta}{8})$, we will not have $p_{\text{est}} \geq p^* - \frac{\Delta_{[2]}}{4}$ via this argument of the max-term.

- For $p_{\mathsf{est}} \geq p^* - \frac{\Delta_{[2]}}{4}$ to be updated by the second argument of max-term, we must have $p_{\mathsf{est}} \geq p^* - \frac{\Delta_{[2]}}{4} - \frac{\Delta_{[2]}}{20} \geq p^* - \frac{\Delta_{[2]}}{2}$ *before* the update. However, Claim D.6 show that before $\mathsf{arm}^*$ arrives, the $\mathsf{king}$ with this large value of $p_{\mathsf{est}}$ will never exhausts its budget on any set of arriving arms $(1 - \frac{\delta}{8})$ and thus will not be discarded; similarly, by Proposition A.2, the $\mathsf{king}$ will not be discarded by the early arms with probability $(1 - \frac{\delta}{8})$. As such the update will not able to happen due to this case either.

Hence, with probability at least $\left(1 - \frac{\delta}{4}\right)$, before the arrival of $\mathsf{arm}^*$, we have $p_{\mathsf{est}} \leq p^* - \frac{\Delta_{[2]}}{4}$.

Conditioning on the above event, now consider the moment when $\mathsf{arm}^*$ arrives. If it arrives as one of the early arms of the epoch, then by Proposition A.4, it becomes the $\mathsf{king}$ with probability at least $(1 - \frac{\delta}{8})$. On the other hand, if it arrives as one of the late arms of the epoch, we show that it is not likely to be defeated at any level of the challenging rule, so the challenge will continue until $\mathsf{arm}^*$ exhausts the budget of the $\mathsf{king}$. More formally, by Proposition A.4,

$$\Pr\left(\text{challenger } \mathsf{arm}^*\text{is defeated at level } \ell\right) = \Pr\left(\widehat{p}_{\text{challenger}} < p_{\mathsf{est}} - \frac{\tilde{\Delta}_\ell}{4}\right) \leq 2\exp(-\log(\frac{1}{\delta_\ell})) \leq 2\delta_\ell.$$

Therefore, by a union bound over all the levels, we have $\Pr\left(\mathsf{arm}^* \text{ ever defeated}\right) \leq 2 \cdot \sum_{\ell=1}^{\infty} \frac{\delta}{50\ell^3} \leq \frac{\delta}{4}$. Finally, a union bound gives us at least $(1 - \frac{\delta}{2})$ probability for both of the above events to hold, concluding the proof. $\qquad\square$

**Lemma D.8** (**Completeness**). *W.p. at least* $(1 - \frac{\delta}{2})$, $\mathsf{arm}^*$ *is not discarded if it becomes the* $\mathsf{king}$.

*Proof.* For the early arms, by Proposition A.4, if $\mathsf{arm}^*$ is already the $\mathsf{king}$, the algorithm does not discard it with probability at least $(1 - \delta') = (1 - \frac{\delta}{8})$.

For the late arms, note that after $\mathsf{arm}^*$ becomes the $\mathsf{king}$, we have $p_{\mathsf{est}} \geq p^* - \frac{\Delta_{[2]}}{2}$ with probability $(1 - \frac{\delta}{8})$ by Line (1) and Proposition A.2. Moreover, $\mathsf{arm}^*$ is no longer part of the stream and hence, by applying Claim D.6, we can show that with probability at least $\left(1 - \frac{\delta}{8}\right)$, the budget of $\mathsf{arm}^*$ never gets exhausted, and therefore it does not get discarded. A union bound among the above cases concludes the proof. $\qquad\square$

A union bound on Lemmas D.8 and D.7 proves Lemma D.2, concluding the proof of Theorem 5.

**Remark D.9.** From the analysis, one can observe that the large constant of the exponent for $\frac{1}{\Delta_{[2]}}$ on the additive sample term (i.e. $\frac{1}{\Delta_{[2]}^7}$) is mainly due to the analysis of Claim D.5, where a large $m_{\mathsf{early}}$ is necessary. We remark that $(i)$. we never need more than $\Theta\left(\frac{n}{\Delta_{[2]}^2}\right)$ sample complexity by running INSTANCE-SENSITIVE-GAME-OF-ARMS, as we can always compute the additive term with $m_{\mathsf{early}}$ and run GAME-OF-COINS if the additive term is too large. In fact, if $m_{\mathsf{early}} \geq n$, the algorithm automatically runs GAME-OF-COINS and outputs its answer. $(ii)$. The analysis of Claim D.5 assumes the worst-case, and the large $m_{\mathsf{early}}$ is necessary only when most $\Delta_{[i]}$'s are close to $\Delta_{[2]}$. It is an interesting future direction to explore how large $m_{\mathsf{early}}$ is required in practical settings.