# OpenReview forum: "Single-pass Streaming Lower Bounds for Multi-armed Bandits Exploration with Instance-sensitive Sample Complexity"
_NeurIPS.cc/2022/Conference — NeurIPS 2022 Accept_

### Official Review · Reviewer_MeNT · 2022-07-10

**Rating:** 6
**Confidence:** 3
**Soundness:** 4 excellent
**Presentation:** 2 fair
**Contribution:** 2 fair

**Summary:**

This paper tackles a recently proposed streaming multi-armed bandit problem with limited memory. They construct an instance-sensitive algorithm when given additional information as input, and provide lower bounds showing that additional information is necessary to achieve good performance with sublinear memory.

**Questions:**

1. The lower bounds indicate classes of instances, to be contrasted with other bandit information theoretic lower-bounds which hold on an instance by instance basis. What are these classes of instances, and can this be generalized?
2. The authors claim that if an $O(1)$-approximation of $H_2^{\delta}$ is given, then the algorithm can still be run successfully. This does not seem to make sense, as if this quantity is rescaled unbeknownst to the algorithm, then it seems as though it will not be able to maintain $\delta$ correctness. Further explanation (and explicitly showing where this parameter is used in the main algorithm) would be helpful.

**Limitations:**

Yes.

Post-discussion: the authors have provided further clarification regarding how the algorithm functions, the motivating setting, and the dependence on $\Delta_{[2]}^{-7}$ (in addition to potential ways that improve it). Incorporating these changes will greatly strengthen this paper in my eyes, and so I am increasing my score by 1 to 6 (weak accept).

**Strengths And Weaknesses:**

Significance: the paper does not seem to be of practical utility, as it seems unreasonable to assume knowledge of either $\Delta_2$ or INST-complexity. This paper appears to have a theoretically significant results however, in the form of the arm-trapping Lemma.

Strengths:
- Arm trapping Lemma is strong, and formalizes intuition regarding the streaming setting.
- Originality: instance-sensitive sample complexity for this setting is new.
- Quality: this work formalizes intuition regarding the difficulty of this problem without additional information.
- Clarity: the paper is written very clearly.

Weaknesses:
- The $\Delta_2^{-7}$ dependence in Theorem 5 seems artifical and quite bad. If we consider problem settings where the minimal gap is scaling as $1/\sqrt(n)$, as is commonly done when considering worst-case scenarios, this term dominates, yielding a sample complexity of $n^{5/2} \Delta_2^{-2}$, dramatically worse than the $n\Delta_2^{-2}$ of the minimax algorithm.
Even for less aggressive choices of $\Delta_2$, it seems as though this will still be a bottleneck (for any $\Delta_2<n^{-1/5}$). Is this fundamental, or can this be improved?
- Almost 2 pages are devoted to proving Lemma 3.1 in the main text; this doesn't help improve intuition, and would be better suited for the appendix. The authors could then present and discuss the algorithms in the main text.
- The algorithms, which are relegated to the appendix, require further explanation; for example, what does the main algorithm take as input? How does it use the input parameter INST-complexity?
- Finally, this work does not seem to be a good fit for NeurIPS. This paper tackles the problem of multi-armed bandits from a theoretical computer science perspective, as can be seen by the key papers this work builds on, which are from COLT and STOC. As this is a subjective point, I am not reducing my score because of it, but I think it merits discussion.

---

> ### Author Response · Authors · 2022-08-02
> **Responses to Questions and Concerns**
>
> Thank you for your constructive evaluations and insightful concerns. Our responses to the questions are as follows:
> - **The exponent on $1/\Delta_{[2]}$ is too large.** Thanks very much for pointing this out. We agree the exponent on $\Delta_{[2]}$ is indeed large in our algorithms. This is mainly due to the very large m_early, which in turn comes from the analysis of Claim D.5, where we used Hoeffding’s inequality with parameter $\theta \approx O(1/\Delta^2_{[2]})$. We believe this can be improved by more careful analysis. Also, we want to remark that for a constant success probability, and assuming $H_2$ is at least $100\cdot n$ (which seems natural in practice), then we only need $10^5$ to $10^7$ arms for the value of $\Delta_{[2]}$ to be around 0.1 ~ 0.2, which is the reasonable number for streaming algorithms. Finally, we plan to conduct experiments in the future to see how large m_early really needs to be.
> - **More space and explanations for the algorithm in the main text.** Thanks a lot for the comment. We appreciate it and we agree that it is a good idea to put more intuitions of the algorithm into the main body and push the proof of lemma 3.1 to the appendix. We will do this for future versions.
> - **General instances vs. adversarial instances.** Thanks for pointing out the concern. Indeed, our lower bounds only work for certain families of instances. Since we need to work on a sample-space trade-off in the streaming setting, we inevitably have to work on adversarial arm arrival. This is in contrast with the information-theoretic lower bounds in the classical settings, where the only concern is the sample complexity. We will add some discussions about this.
> - **$O(1)$-approximation of $H^\delta_2$.** We agree that this expression is imprecise. We meant to say that if there exists a fixed constant $c$ such that the knowledge is $H^\delta_2$, then our algorithm still works. We agree that if $c$ is chosen by an adaptive adversary, then the algorithm breaks. We will clarify this in future versions.
> - **Fitness to the scope of the conference.** Thanks for pointing out the concern. We want to remark that while the line of work is built on a STOC paper, many related papers did appear at venues like ICML (e.g. Karnin et al. [ICML 2013]; Jin et al., [ICML 2021]) and NeuIPS (e.g. Maiti et al., [NeuIPS 2021]).

---

> > ### Comment · Reviewer_MeNT · 2022-08-08
> > **Questions regarding $\Delta_{[2]}$ dependence and motivating example**
> >
> > - Regarding the authors discussion with reviewer LGML: H2 and H1 are bounded by each other up to a log(n) factor. So, knowing H_1, can the proposed algorithm not simply be run with $\hat{H}_2 = \log n H_1$?
> > - I still do not fully understand how the algorithm proceeds, e.g. what its inputs are, and how it uses the parameter INST-complexity. It would have been nice to see a revised version of the manuscript during this period.
> > - What practical setting are the authors considering where simultaneously a) there are many (e.g. 10^7 arms), b) $\Delta_{[2]}$ is known or lower bounded, c) $H_2$ is known or upper bounded, and d) the arms are arriving in a random order?
> > - Do the authors see a potential path towards improving the dependence on $\Delta_{[2]}$? Is this perhaps improvable if a larger constant memory is allowed?
> > - Scope of conference: NeurIPS is indeed a broad venue, and no other reviewers seem to take issue with the fit, so I’ll drop this point.
> >
> > Overall the paper is well written, and the provided lower bound (and accompanying lemma) is novel and clearly formalizes the hardness of the problem at hand. However, both the description / formalization of the proposed algorithm and its motivation / justification currently seem to be lacking. Thus, I am maintaining my score of borderline accept.

---

> > > ### Author Response · Authors · 2022-08-09
> > > **Responses to the Additional Questions and Concerns**
> > >
> > > Thank you for the additional questions and concerns. For the purpose of clarification, we want to add the following comments (We want to emphasize that these responses serve the purpose to clarify. We completely understand and respect your decision to not update the score.):
> > > - **No revised version of the paper.** While NeurIPS allows revised versions of the paper to be submitted during the discussion period, we believe that the time is not sufficient to carefully revise the manuscript to reflect the reviewers’ comments. In particular, thanks to the reviewers’ comments that helped us find better ways to frame our theorems, we want to take the time to update a complete version, as opposed to only changing the algorithm part and submitting a semi-baked one. Furthermore, since there’s no factual error to correct in our manuscript, we believe a rushed revised version is not necessary.
> > > - **Extra $\log(n)$ multiplicative term.** Thanks for the comment. The extra $\log\log(1/\Delta)$ factor is not necessarily bounded by $\log(n)$, as we can easily pick $\Delta_{[2]} = 1/2^{2^n}$. We agree that in most practical settings, $H_1$ and $H_2$ should be within an $O(\log(n))$ factor of each other. However, an $O(\log(n))$ *multiplicative factor on the $\omega(n)$ sample complexity* is considered very large, in the sense that the whole point of Assadi and Wang [STOC 2020] is to shave off an $O(\log(n))$ factor in the sample complexity.
> > > - **How does the algorithm use INST-complexity?** Thanks for the concern. We need to compute the budget that we give to the king (stored arm) with the INST-complexity when 1. The king passes the GAME-OF-COINS procedure with the early arms; and 2. The king has a new challenger from the late arms. The budget idea was first introduced by Assadi and Wang [STOC 2020]. In their work, since they are only interested in the $O(n/\Delta^2_{[2]})$ sample complexity, they can immediately compute the ‘budget’ after knowing $\Delta_{[2]}$. However, in our settings, we need to know the INST-complexity quantity in advance to compute such a budget. Knowing all the sub-optimal gaps (i.e. ${\Delta_{[i]}}_{i=2}^{n}$) will definitely be sufficient, and we use the INST-complexity which is a weaker condition.
> > > - **The practical settings for the algorithm.** Thanks for pointing out the concern. It is easy for modern applications to have a large number of arms (say, online retailing of large corporations), and the streaming setting is exactly introduced to deal with scenarios when the number of arms is quite large, say, more than $10^7$. A lower bound for $\Delta_{[2]}$ to be 0.1 ~ 0.2 is also realistic – consider finding the daily best seller on a large retail website, and it is common for the best option to be way more popular than even the second best option.  We agree that the random order and the knowledge of INST-complexity are stronger conditions. To the best of our knowledge, the former condition can be met in applications where we have a ‘bank’ that store all the arms and a local machine to read and process them. The second condition becomes possible when one can estimate the INST-complexity budget by historical data.
> > > Finally, we agree that it’s not very common for all the conditions of the algorithm to hold. Our earlier response focuses on the claim that in real-world applications if c) and d) hold, it is realistic to get the INST-complexity as opposed to a sample complexity that is dominated by the additive term. Our central message of the paper is that instance-sensitive exploration in a single-pass stream is *hard*.
> > > - **Potential ways to reduce the exponent on $\Delta_{[2]}$.** Thanks for the question. It is unclear whether increasing the memory to constant is helpful to reduce the exponent on $\Delta_{[2]}$. We need $m_{early}$ to be $\log(n)/\Delta^c_{[2]}$ for some $c\geq 2$, and inside this subroutine, our lower bound implies that it’s not possible to break $\Omega(m_{early}/\Delta^2_{[2]})$ with any memory of $o(m_{early})$. As such, increasing the memory to some larger constant seems not immediately helpful in our algorithm. However, we believe it’s possible to reduce the value of $c$ with better analysis (currently $c=4$).

---

> > > > ### Comment · Reviewer_MeNT · 2022-08-09
> > > > **Response (increasing my score by 1)**
> > > >
> > > > 1. **$H_1$ versus $H_2$**: my mistake, I believe that classically these quantities are defined without the $\log\log(1/\Delta)$ factors (e.g. "Almost Optimal Exploration in Multi-Armed Bandits"), but following the notation in your paper the ratio is indeed not bounded by $\log n$. I believe that achieving instance-sensitive sample complexity is novel and exciting, even if suffers from small polylogarithmic factors.
> > > > 2. **Algorithm:** Thanks for the explanation! I think (as previously discussed) additional exposition regarding the algorithm would be beneficial; it was hard for me to parse the algorithms on their own in the appendix. I think that this succinctly explains why the current approach suffers from such a dependence on $\Delta_{[2]}$, and points towards potential avenues for improving it, which would be helpful to include in the main text.
> > > > 3. **Setting:** The appeal to historical data makes sense for estimation of INST-complexity; I would urge the authors to flesh this setting out a little bit more. I completely agree that the main message of this paper (and its primary technical contribution) are in the lower bounds provided. It just seems that if an achievability scheme is provided, justification regarding the reasonability of its assumptions and an example of a setting where it could be run should also be provided.
> > > >
> > > > Final additional question: if the input $\hat{\Delta}\_{[2]}$ is overestimated, i.e. $\hat{\Delta}\_{[2]}>\Delta_{[2]}$, but all other assumptions hold, can the proposed algorithm guarantee that a $\hat{\Delta}\_{[2]}$ optimal arm is found?

---

> > > > > ### Author Response · Authors · 2022-08-10
> > > > > **Thanks for your responses and answer to the additional question**
> > > > >
> > > > > Thanks for the additional comments and suggestions. Regarding the last question, we don't think the algorithm works with an overestimated $\hat{\Delta}$. This setting is called the $\epsilon$-best arm problem in the literature (where $\epsilon$ is the suggested $\hat{\Delta}$ parameter -- only a change of notation), and the goal is to find the arm with a reward that is $\epsilon$-close to the best arm. The problem is interesting and it is studied on its own in related work.
> > > > >
> > > > > Even with the sample complexity of $O(n/\epsilon^2)$, the memory complexity in Assadi and Wang [STOC 2020] is already $\log^{*}(n)$ (as opposed to 1 arm with an accurate $\Delta_{[2]}$). The gap was later closed by Jin et al. [ICML 2021], which managed to reduce the memory to 1 arm. However, it is unclear how their algorithm can be used in ours, and the notion of using the $H_2$ complexity with an estimated $\hat{\Delta}$ seems not well-defined (i.e. if we should count for the complexity induced by the gaps that are smaller than $\hat{\Delta}$ ($\epsilon$)).
> > > > >
> > > > > Here's a short analysis of why the algorithm in our paper doesn't work with an overestimated $\hat{\Delta}$: even in the GAME-OF-COINS subroutine, the best arm may be lost at some point. Say that after losing the best arm, the k-th best arm is now the king, and $\Delta_{[k]}<\hat{\Delta}$. This seems nice since we still have an arm that is better than the $\hat{\Delta}$ gap. However, note that later arms (say, arm i) will challenge the k-th best arm with gap $\Delta_{[k, i]}:=\hat{\Delta}-\Delta_{[k]}$. This quantity may be much less than $\hat{\Delta}$, so we could wrongly discard the k-th best arm in the procedure. The same argument applies to the arm that attains the $\hat{\Delta}$-gap reward -- we may also wrongly discard it in the stream.
> > > > >
> > > > > We will add some discussion about this in the later versions to help readers understand the difference between the two tasks (finding the best vs. $\epsilon$-best arm). Thanks for this nice question.

---

### Official Review · Reviewer_qt4b · 2022-07-11

**Rating:** 6
**Confidence:** 3
**Soundness:** 3 good
**Presentation:** 4 excellent
**Contribution:** 3 good

**Summary:**

This paper studies the classical multi-armed bandits problem under the streaming setting with the goal of identifying the best arm using a fixed confidence. Unlike the previous work by Assadi and Wang [STOC 2020] which introduces a worst-case optimal algorithm, this paper aims to design a near-optimal instance-sensitive single-pass streaming algorithm which performs better when input problem instance is not hard. The contribution of this paper is three-fold: 1) It shows negative result that it is impossible to achieve such an algorithm even when the smallest gap $\Delta_{[2]}$ is known beforehand. 2) There exists a near-optimal algorithm when $\Delta_{[2]}$ and the complexity of the input instance are known, and the arms arrive completely randomly. 3) A by-product of this paper is the arm-trapping lemma which applies to the goal of outputting $o(n)$ arms that contains the best arm.

**Questions:**

Instead of designing a one-pass streaming algorithm, it might be more interesting to see if multiple passes could help to this problem? Have the author(s) considered such direction?

**Strengths And Weaknesses:**

* Strengths
** The paper is written well and in good presentation which makes it easy to follow. I did not check the proof details in the appendix. But the proofs in the main part look solid and reasonable to me.
** This paper is the first to show impossible result when the goal is to design a near-optimal instance-sensitive single-pass streaming algorithm

* Weaknesses
** The proposed algorithm is not practical since one has to know $\Delta_{[2]}$ and the complexity of the input instance which is hard to know beforehand.

---

> ### Author Response · Authors · 2022-08-02
> **Responses to the Questions and Concerns**
>
> Thank you for your encouraging and insightful evaluations and questions. Our responses to the questions are as follows:
> - **The exponent on $1/\Delta_{[2]}$ is too large.** Thanks very much for pointing this out. We agree the exponent on $\Delta_{[2]}$ is indeed large in our algorithms. This is mainly due to the very large m_early, which in turn comes from the analysis of Claim D.5, where we used Hoeffding’s inequality with parameter $\theta \approx O(1/\Delta^2_{[2]})$. We believe this can be improved by more careful analysis. Also, we want to remark that for a constant success probability, and assuming $H_2$ is at least $100\cdot n$ (which seems natural in practice), then we only need $10^5$ to $10^7$ arms for the value of $\Delta_{[2]}$ to be around 0.1 ~ 0.2, which is the reasonable number for streaming algorithms. Finally, we plan to conduct experiments in the future to see how large m_early really needs to be.
> - **Multi-pass algorithms.** Thanks a lot for the excellent question. Indeed, Jin et al. [ICML 2021] proposed an algorithm with $O(\log(1/\Delta_{[2]}))$ passes without any additional knowledge (even without the knowledge of $\Delta_{[2]}$). On a closely-related problem of multi-round collaborative learning, Tao et al. [FOCS 2019] give a lower bound of ~ $\Omega(\log(1/\Delta_{[2]}))$ *rounds*. However, we realized that their collaborative learning setting cannot be reduced to streaming. As such, the tight pass bound for instance-sensitive sample complexity is a very interesting open problem to explore.

---

### Official Review · Reviewer_dDQ4 · 2022-07-12

**Rating:** 8
**Confidence:** 3
**Soundness:** 4 excellent
**Presentation:** 3 good
**Contribution:** 4 excellent

**Summary:**

This paper studies space vs. sample complexity tradeoffs for stochastic multi-armed bandits. While it is possible to achieve optimal worst-case sample complexity while storing only a single arm in the streaming model, this paper investigates whether the same is true for instance-optimal bounds. The main result is a lower bound, showing that any single pass streaming algorithm must either incur a worse sample bound or store $\Omega(n)$ arms. The authors show that this lower bound holds even under several stronger assumptions. Finally, the authors give an upper bound in the case where the stream is randomly ordered and the algorithm is furnished with some statistics about the problem instance.

**Questions:**

- Is there generally an understanding of the relationship between the arm finding and trapping problems? Is there a regime in which there is a separation between the problems (i.e. trapping is easier than separation)?

**Limitations:**

This paper is removed from direct applications, and I am satisfied with the authors' response.

**Strengths And Weaknesses:**

Strengths
- The main technical tool in the paper is an interesting lemma in its own right which states that in the worst-case, $\Omega(n/\Delta^2_{[2]})$ samples are needed to output a set of $o(n)$ arms which contain the best arm (where $\Delta_{[2]}$ is the gap between the best and second best arms).
- Leveraging this lemma, the paper elegantly shows exactly which pieces of extra information/conditions on the stream are needed to circumvent this lower bound.   For instance, prior works assumed knowledge of $\Delta_{[2]}$, but this paper shows that without such knowledge, any algorithm with $o(n)$ memory requires arbitrarily high sample access. As a result, the paper gives a very clear (and somewhat surprising) picture of memory/sample tradeoffs for this problem.

Weaknesses
- I think the paper would benefit from including some text explaining how the main results connect in the overview. In particular, explaining how the arm trapping lemma (which describes a worst-case distribution) can be applied to give lower bounds in the instance-optimal regime earlier on would be helpful.

---

> ### Author Response · Authors · 2022-08-02
> **Explanations and responses to the questions**
>
> Thank you for your positive feedback and the helpful questions. Our responses to the questions are as follows:
> - **More high-level explanations for how the arm-trapping lemma is used in the lower bounds.** Thanks very much for pointing this out. We agree that such a high-level explanation will help readers to understand how the arm-trapping lemma works in the lower bounds. We will add the discussions in the future version.
> - **The study of arm trapping vs. identification. Thanks for raising this insightful question.** To the best of our knowledge, there have not been any results studying the separation between trapping and identification. Indeed, one of the reasons for us to find the results interesting is precisely because it shows under certain conditions, trapping is as hard as identification. More insights on this problem are definitely an interesting direction for future explorations – and we appreciate the question.

---

### Official Review · Reviewer_LGML · 2022-07-14

**Rating:** 7
**Confidence:** 3
**Soundness:** 3 good
**Presentation:** 4 excellent
**Contribution:** 4 excellent

**Summary:**

This paper considers the tradeoffs between samples (i.e., number of arm-pulls) and space (i.e., maximum number of arms kept in memory) in the pure exploration stochastic multi-armed bandit problem. The objective in their MAB setting is to output, with a fixed probability, the arm with the highest mean reward, while minimizing the sample complexity (i.e., the total number of arm pulls), and while storing as few arms in memory as possible. They prove that any algorithm which stores $o(n)$ arms must have sample complexity that is $\omega(\text{INST-complexity})$, unless the arms arrive in uniformly random order and the algorithm knows the $\text{INST-complexity}^*$. Further, any algorithm which stores $o(n)$ arms has sample complexity which is unbounded as a function of $n$ and $\Delta_{[2]}$, unless some parameter of the problem (e.g., $\Delta_{[2]}$) is known. They complement this result by proving that, assuming the INST-complexity and $\Delta_{[2]}$ are known to the algorithm, and the arms arrive in a uniformly random order, then there is an algorithm which uses the memory of a single arm, and has sample complexity which nearly matches the best-known upper-bound (achievable with an arm memory of $n$).

$*\text{While}$ this is the claim of the paper, I _think_ the actual result requires weaker knowledge on the suboptimality gaps than INST-complexity. More on this later.

**Questions:**

Many of my concerns are listed in the **Weaknesses** section above. My main questions (some not listed in that section above) are as follows:

- It may be useful to formalize what it means for an algorithm to not know a particular parameter. If a parameter is known only up to a $(1+\epsilon)$ multiplicative factor, for instance, what is the smallest value of $\epsilon$ which constitutes “not knowing”?
- It seems that Theorem 4 does not require knowledge of INST-complexity exactly, but some function on the suboptimality gaps (e.g., $H_1$) also seems sufficient. Could you characterize exactly what information is needed for your lower bounds to still hold? Can you rephrase Result 1(i) to be more accurate (as discussed above)?
- On the upper bound side – if the algorithm instead knew $H_1$ instead of $H_2^\delta$, would your sample complexity results still hold? It would be interesting to me to see if the proof breaks down, or not.
- Regarding the random arrival order – could you formalize exactly how random the arrival order needs to be? I suppose a uniformly-random arrival order works, but what if the arrival order is slightly biased? It seems if this bias is small, then your lower bound techniques likely would still work. Is there hope for a guarantee for arrival order distribution that is at most $\epsilon$-far away (in TV distance, for example) from the uniform distribution?
- You mention on line 41 that the goal is to identify the best arm with high _constant_ probability. Is there any hope for similar results that hold, e.g., with probability $1/n$?
- Do you think there is any hope to show a lower bound on the necessity of the second-order terms in your sample complexity upper bound? Or do you think that these could be removed with a more refined analysis?
- A question about the streaming setting you consider – if I understand correctly, your model constrains the memory in terms of the number of arms that can be pulled at any time, but not on the statistics that you can maintain about previous arms. So, for instance, an algorithm which maintained statistics on all $n$ arms (thus, in a different sense, using $O(n)$ memory), could still be considered “sublinear memory” in your model, right? Can you give any motivation for this setting, where you are constrained only on the number of arms you can pull simultaneously? Is this the most natural streaming model one could consider for your problem setting? Or is constraining the memory used for statistics also interesting?


**Strengths And Weaknesses:**

**Strengths**

This paper makes significant progress on an interesting and natural problem of the sample complexity of limited-memory pure exploration problems. The lower bounds show that instance-sensitive algorithms with the correct sample complexity are not achievable without some side information provided to the algorithm. Their bounds give some evidence for what side information may be necessary to solve the problem. On the upper-bound side, they provide an algorithm which achieves a near-optimal sample complexity with the memory of only a single arm, given the side information of $\Delta_{[2]}$, $H_2$, and a uniformly random arrival order. The proof here appears quite interesting, and it seems an interesting extension of the ideas used in [4].

In general, I think the authors do a nice job of explaining and giving intuition in a simple manner for all of their results.

**Weaknesses**

I think the theorem statements for some of the lower bounds are presented a bit too informally. In particular, I think it would be helpful to formally define what is meant by an algorithm not knowing parameters such as $\Delta_{[2]}$ or $H_2$. For instance, if the algorithm does know $\Delta_{[2]}$ exactly, but knows this parameter to a $(1+\epsilon)$ multiplicative factor for small $\epsilon$, I guess this is still sufficient, right? Perhaps it would be good to specify how large $\epsilon$ can be to still achieve your results. Further, if I understand it correctly, it seems that knowledge of $H_2$ is not really necessary for Theorem 4 to hold. For example, it seems like the lower bound would also be true if instead, the algorithm did not know $H_2$, but knew $H_1$ instead. In fact, I think this lower bound is true as long as the algorithm knows some value $f(\\{\Delta_i\\}_{i\neq i^*})$ (perhaps with some constraints on how perturbing a $\Delta_i$ affects the function value), right? So it is not clear to me what information about the instance should actually be necessary to obtain near-optimal sample complexity with $o(n)$ arm storage.

So, in this sense, Result 1 (i) on line 99 is not quite correct, right? (I realize that this is meant to be an informal statement, but perhaps it would be better to reword it to be more accurate). Since if INST-complexity is not known, but, say, $H_1$ is known instead, then it is possible to have $O(H_2)$ regret, right (assuming random arrival orders and knowledge of $\Delta_{[2]}$).

Another (very slight) weakness of the lower bounds result is that the meaning of “instance-sensitive” seems slightly different than is standard in multi-arm bandit literature (at least, as I understand it – please point out if I am mistaken). The style of instance-dependent lower bounds I am familiar with holds for _any_ instance with suboptimality gaps $\\{\Delta_i\\}\_{i\in[n]}$. See, for example, Lemma 16.1 from “Bandit Algorithms”,  Lattimore and Szepesvári 2020. By contrast, I think your lower bounds are _slightly_ weaker than this, in the sense that they only hold for _some_ environment which has suboptimality gaps $\\{\Delta_i\\}_{i\in [n]}$. Perhaps it would be good to make this distinction a bit more clear somewhere in your paper.

Finally, I find it somewhat unfortunate that the streaming algorithm you provide gets such little attention in the main body of the paper. I realize that you have very limited space, and the lower bound techniques are cool! But perhaps there is a way to discuss this a bit more in the main body, since the upper bound techniques also seem quite interesting! One possible idea is to cut down on the discussion of the arm trapping lemma proof – perhaps this can be just a sketch, and move the full proof to the appendix?

---

> ### Author Response · Authors · 2022-08-02
> **Responses to the comments and questions -- Part I**
>
> (The responses are in two parts due to character limits in each comment)
>
> Thank you for your careful evaluation, positive feedback, and insightful questions. We believe with your questions and observations, the quality of the paper can be greatly improved. Our responses to the questions are as follows:
> - **The lower bound with approximate knowledge of $\Delta_{[2]}$.** Thanks for pointing out the interesting question. Indeed, we can establish a trade-off between the precision of $\Delta_{[2]}$ and the sample complexity. This result will stand between the case that we have no knowledge of $\Delta_{[2]}$ (which we proved the sample complexity is unbounded with $o(n)$ memory) and the upper bound of $O(n/\Delta_{[2]}^2)$ with 1 arm when $\Delta_{[2]}$ is known. In particular, given an $\alpha$-approximation of $\Delta_{[2]}$, we can show that an $\Omega(\alpha^2 \cdot n/\Delta_{[2]}^2)$ sample complexity is necessary unless using $\Omega(n)$ memory; furthermore, the algorithm of Assadi and Wang [STOC 2020] would similarly run in $O(\alpha^2 \cdot n/\Delta_{[2]}^2)$ arm pulls with a memory of a single arm. We will add some discussion for this result in future versions.
> - **The upper and lower bounds with the knowledge of $H_1$ and the sub-optimal gaps $\{\Delta_{[i]}\}_{i=2}^{n}$.** Thanks very much for bringing up the excellent observation. We checked the hard distribution for our proof for the $H_2$ instance again and realized that it works even with the knowledge for every $\Delta_{[i]}$. We appreciate this nice observation as both the knowledge of $H_1$ and $H_2$ are somehow weaker forms of the knowledge of $\Delta_{[i]}$'s. As such, we will change the description of the theorem in future versions.  As for a more detailed discussion of $H_1$: we did not consider $H_1$ as one of the candidate parameters in our lower bounds. To the best of our knowledge, it is not known whether $H_1$ is sufficient for best-arm exploration even in the classical setting (i.e. n-arm memory). In Jamieson et.al. [COLT 2014], it is shown that the $H_2$ ($=\omega(H_1)$) sample complexity is necessary for the case of 2 arms. Later, Chen and Li [COLT 2016] show that the $H_1$ sample complexity is possible (albeit with $\log\log$ term for $\Delta_{[2]}$) for some special instances, and they conjectured a lower bound that will rule out $H_1$ complexity for general n. This is the state-of-the-affair for classical sample complexity as far as we know. As such, it is unclear whether the knowledge of $H_1$ is sufficient for any upper bound result.
> As for our algorithm: the solution of Karnin et al. [ICML 2013], which we are `simulating’ in the streaming settings, inherently requires the $H_2$ sample complexity (to estimate the unknown $\Delta_{[i]}$). In our algorithm, if we only know the quantity of $H_1$, we will risk wrongly losing the king even though it is the best arm due to budget exhaustion.
> - **Parameter knowledge settings in the lower bounds.** We will improve the theorem statement in the future version. Specifically, thanks to your insightful comments, we can now change the statement of $H_2$ to sub-optimality gaps, and we will explicitly state what are the conditions/knowledge we are using in our lower bounds.
> - **Questions about the random order arrival.** We confirm that by saying random order we mean the arrival order to be uniformly at random, and we will add a formal explanation in the future version. The order that is $\epsilon$-far from random sounds very interesting. In general, this only makes the instances more adversarial and the lower bound should still hold. It is interesting to see how far from truly random can our algorithm tolerate, and that will be a direction to explore in the future.
> - **Question about the high constant probability.** Thanks for raising the question. A $1-1/n$ probability is impossible with the current regime of sample complexities in the following sense: consider two arms with mean rewards $p$ and $p+\Delta$, respectively. Now to distinguish the two arms w.p. $1-1/n$, it is necessary and sufficient to use $O(\log(n)/\Delta^2)$ arm pulls (up to lower order terms of n since Chernoff bound is almost tight). As such, if the sample complexity is independent of extra multiplicative terms of $\approx \log(n)$, it’s not possible to get a $1-1/n$ success probability.

---

> > ### Author Response · Authors · 2022-08-02
> > **Responses to the comments and questions  -- Part II**
> >
> > (continued due to character limits)
> > - **Question about the additive term in the algorithm.**  This is indeed a very interesting question and we do not know the answer. It seems promising to decrease the exponent of $\Delta_{[2]}$ by using some tighter analysis (currently we are canceling the upper bound of the individual random variables in Hoeffding’s inequality ($\theta = 1/\Delta^2_{[i]}$) with m_early = $O(1/\Delta^4_{[2]})$ in claim D.5). However, exploring whether the additive term is necessary is indeed an open problem for our next steps. We suspect that some dependence on $\Delta_{[2]}$ beyond what is needed in the non-streaming setting should be required here.
> > - **Question about storing extra statistics.** Thanks a lot for the question. The model initiated by Assadi and Wang [STOC 2020] defined the space complexity of streaming MABs as the maximum number of arms to be stored. As such, in the algorithms, we simply follow their model which is accepted in the area. Furthermore, we know that in related models such as submodular maximization, storing items (in this case arms) can qualitatively be costlier than storing statistics. Note however that our algorithms only require storing statistics proportional to their memory, thus only $O(1)$ additional machine words, and as such are quite efficient in that regard as well. On the other hand, the lower bounds work even in this more relaxed version that only limits the number of arms stored by the algorithm and allows for storing arbitrary additional statistics for free.
> > - **The usage of the term ‘instance-sensitive’.** Thanks for pointing out the discrepancy in the terms. We realize that there are different terms the community has used for the instance-dependent bounds (e.g. earlier work like Karnin et al. [ICML 2013] and Jamieson et.al. [COLT 2014] simply call this bound (almost) optimal; Chen and Li [COLT 2016] called it instance-wise optimal, which was followed by the open problems posed by Assadi and Wang [STOC 2020]). We realized that it is unclear whether the $\log\log(1/\Delta_{[i]})$ term is necessary for optimal bounds, so we call it ‘instance-sensitive’ as opposed to instance-optimal. We will add more discussions to clarify this for the future version.
> > - **More space for the algorithm in the main text.** We appreciate the suggestion and we too agree that it is a good idea to put more intuitions of the algorithm into the main body and push the proof of lemma 3.1 to the appendix. We will do this for future versions.

---

> > > ### Comment · Reviewer_LGML · 2022-08-08
> > > **Follow-up to Author response**
> > >
> > > Thank you for your detailed response to my questions. After reading the response, as well as the questions/concerns of other reviewers, I stand by my original evaluation of this paper. I do not see the concerns of the other reviewers as sufficient reason to change my score. I would definitely encourage the authors to update their paper with some of the discussion above, particularly in regards to the formalism of the lower bounds, as I think that they have found a nicer way of stating precisely the restrictions on the classes of algorithms for which the lower bounds hold.

---

### Meta-Review · Area_Chair_Sskm · 2022-08-20

**Recommendation:** Accept
**Confidence:** Certain

**Metareview:**

This paper studies the pure exploration in the streaming MAB model.  The main message it delivers is that any single pass streaming algorithm must either have a large sample complexity or store a linear number of arms.  The reviewers agreed that the results are interesting and that the analysis in the paper is novel and technically strong.  Some questions were raised regarding the requirement of knowing Delta_[2] in the algorithms and the high dependence on 1/Delta_[2] in the upper bounds. These questions have been largely addressed by the authors' responses.  There are still some questions regarding the motivation of the streaming bandit model; it will be nice if the authors can give some real-world applications for this model in the next version.

**Award:**

No

---

### Decision · Program_Chairs · 2022-09-14

Accept